# Combinatorial entropy behaviour leads to range selective binding in ligand-receptor interactions

Meng Liu [1,2,11], Azzurra Apriceno [3,4,11], Miguel Sipin[3,4], Edoardo Scarpa [3,4], Laura Rodriguez-Arco [3,4], Alessandro Poma [5], Gabriele Marchello[4,6,7], Giuseppe Battaglia [3,4,7,8,9] & Stefano Angioletti-Uberti [2,10]

From viruses to nanoparticles, constructs functionalized with multiple ligands display peculiar binding properties that only arise from multivalent effects. Using statistical mechanical modelling, we describe here how multivalency can be exploited to achieve what we dub range selectivity, that is, binding only to targets bearing a number of receptors within a specified range. We use our model to characterise the region in parameter space where one can expect range selective targeting to occur, and provide experimental support for this phenomenon. Overall, range selectivity represents a potential path to increase the targeting selectivity of multivalent constructs.

[1] Beijing Advanced Innovation Centre for Soft Matter Science and Engineering, Beijing University of Chemical Technology, Beijing, People's Republic of China. [2] Institute of Physics, Chinese Academy of Science, Beijing, People's Republic of China. [3] Department of Chemistry, University College London, London, UK. [4] Institute for the Physics of Living Systems, University College London, London, UK. [5] Division of Biomaterials and Tissue Engineering, Eastman Dental Institute, University College London, London, UK. [6] Physical Chemistry Chemical Physics Division, Department of Chemistry, University College London, London, UK. [7] The UCL EPSRC/JEOL Centre for Liquid Phase Electron Microscopy, London, UK. [8] Institute for Bioengineering of Catalonia (IBEC), The Barcelona Institute for Science and Technology (BIST), Barcelona, Spain. [9] Catalan Institution for Research and Advanced Studies (ICREA), Barcelona, Spain. [10] Department of Materials, Imperial College London, London, UK. [11] These authors contributed equally: Meng Liu, Azzurra Apriceno. ✉email: g.battaglia@ucl.ac.uk; sangiole@imperial.ac.uk

In nature, binding occurs with an exquisite selectivity that we are still striving to achieve in synthetic systems. For example, some viruses can attach to specific cell types without infecting others, a mechanism that is already being exploited for the development of more selective cancer therapy[1]. Similarly, antibodies recognise, i.e. bind, particular epitopes with very high strengths, yet tiny molecular-level variations can make them completely ineffective, which is why every year we need to develop new vaccines against influenza, for example. In many cases, binding in these biological entities occurs by the formation of multiple bonds between their ligands and complementary receptors on the target, typically referred to as multivalent binding.

That nature uses this binding modality to achieve high selectivity should not come as a surprise. In fact, various studies have unravelled the way binding selectivity can be enhanced by multivalency[2–11]. In particular, in the last decade so-called multivalent super-selectivity has arisen as a hot topic for the development of targeted drug delivery as well as biosensing[12]. More precisely, super-selectivity refers to the ability of multivalent construct to have a much sharper response to gradients in receptor density compared to monovalent ones and can be used to obtain an (almost) perfect on-off behaviour, where binding occurs exclusively above a certain number of receptors. One of the earliest, if not the earliest, experimental proofs of this concept was given in the seminal paper of Carlson et al.[2], which showed that cancer cells overexpressing receptors, a typical occurrence in various types of malignancies, can be better discriminated compared to healthy ones using multivalent rather than monovalent binding. In 2011, the microscopic origins of this behaviour have been explained by Martinez–Veracoecha and Frenkel[6], using an analysis rooted in statistical mechanics that highlighted the importance of the combinatorial binding entropy due to the various binding patterns achievable when multiple ligands and receptors are present. These results have now been validated several times, both by Monte Carlo calculations as well as by experimental data on different multivalent systems, thereby highlighting their generality[2,4–6,9–11].

In recent years, also thanks to advances in formulating a general theory of ligand–receptor mediated interactions[13–18], we have been able to uncover other potential benefits of multivalent targeting, as well as drawbacks[7], considering more general scenarios including multiple receptor types[19] and the effect of spurious, off-target interactions[7]. In this article, using a combination of theory, numerical modelling and experiments, we present a qualitatively different type of selective targeting which arises in systems where attraction is dominated by the formation of ligand–receptor bonds: the ability, under appropriate conditions, to only bind to targets where the receptor density is within a certain range, but not below nor above. We dub this phenomenon range selectivity.

## Results

### The basic theoretical model.
As an archetypal example, we consider here the case of a solution containing multivalent nanoparticles that can adsorb on a surface. The nanoparticles are coated by both targeting ligands complementary to the receptors on the surface and a protective polymer brush, e.g. poly-ethylene glycol (see Fig. 1). This is a design commonly found in nanocarriers for drug delivery, where the polymer brush is mainly used to avoid protein adsorption[9,20]. This latter circumstance, in fact, can lead to either removal of the nanoparticle from the blood stream, an immune reaction or simply loss of targeting by shielding the ligands[21]. In the system depicted in Fig. 1, nanoparticle adsorption is driven by the formation of bonds between

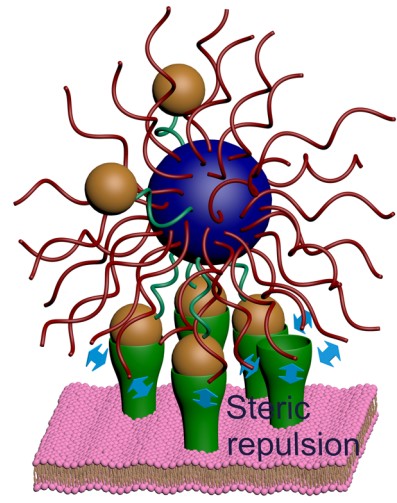

**Fig. 1 Schematic representation of our system.** A nanoparticle coated with ligands (green tethers and orange spheres) interacts with a receptor-coated surface (green funnels). The attractive interaction arises from the formation of ligand–receptor bonds. The presence of excluded volume interactions (here schematically represented as blue arrows), e.g. due to interaction between the receptor and the nanoparticle coating (red tethers), or between the ligands and the grafting surface of the cell (here shown as a lipid bilayer), provides an additional repulsive interaction. Crucially, the scaling of the two with respect to the number of ligands and receptors is different, giving rise to what we dub here range selectivity.

its ligands and receptors on the surface. The simple question we ask is the following: how does the probability of the nanoparticle binding to the surface change as a function of the number of receptors? As previously shown[6,7,10], this adsorption probability $\theta$ can be described via a Langmuir-like expression:

$$\theta = \left\langle \frac{zq(N_L, N_R, \beta\Delta G)}{1 + zq(N_L, N_R, \beta\Delta G)} \right\rangle_{N_X}. \tag{1}$$

where $N_{X=L,R}$ is the number of ligands and receptors, respectively, and $\Delta G$ is the bond free-energy. Throughout the paper, $\beta = k_B T^{-1}$, where $k_B$ is Boltzmann's constant and $T$ temperature, hence $\beta$ is the inverse thermal energy. With the angle brackets $\langle \rangle$ we indicate an average over a Poisson distribution. This average is taken to account for inhomogeneities in the spatial distribution of receptors and / or ligands, which can be related either to the grafting procedure or to binder mobility on the surface. It should be noted that the exact form of this distribution is not important for the appearance of the effects we describe. In fact, the same trends are observed if using a Gaussian rather than a Poisson distribution, or even without any averaging at all. Finally, $z$ is the nanoparticles' activity in the bulk solution, which for homogeneously coated nanoparticles and dilute solutions can be taken equal to their number density[6].

The central quantity in Eq. (1) to describe this problem is $q$, the partition function of the nanoparticle in the bound state, which depends on the number of ligands and receptors available for binding, as well as on the strength of their bond. This partition function can be written as[10] $q = v_{bind} \exp(-\beta F_{tot})$ where $v_{bind} = \pi R^2 L$ is the binding volume of the adsorption site, $R$ being the radius of the nanoparticle (including any contribution from an eventual protective polymer coating) and $L$ the range of distances at which the particle can form bonds, which we can set to be equal to roughly the gyration radius of the ligand's tether $R_g$ (see the Supplementary Notes I), and $F_{tot} = F_{att} + F_{rep}$ is the free-energy of adsorption.

**The attractive and repulsive contributions of ligands and receptors**. In this system, there are two contributions to $F_{tot}$. On the one side, we have an attractive contribution $F_{att}$, generated by the formation of ligand–receptor bonds. On the other side, we must consider that both receptors and ligands also provide a repulsion $F_{rep}$, due to the excluded volume interactions that arise in the crowded environment of the binding region (see Fig. 1 and Supplementary Fig. 6 in the Supplementary Notes I for clarity). For example, in the typical case of polymer-coated nanoparticles approaching the cell surface, receptors will feel the repulsion due to compression of the polymer brush upon binding[9]. Similarly, ligands can feel excluded volume interactions due to the cell glycocalyx, the ubiquitous polymer layer present on the surface of cells[22], as well as due to the cell membrane on which the glyco-calyx is grafted.

Besides the single-bond energy, the attractive part $F_{att}$ crucially depends on the number of binding configurations available[13,14], which, in turn, depends on the exact spatial distribution of both ligands and receptors. What is important to show our point is that the magnitude of this contribution is bound between a lower and an upper value, given by the following formulas, respectively[17]:

$$\beta F_{att} = -\ln\left[1 + N_R N_L \exp(-\beta \Delta G)\right] \approx -\ln(N_R) - \ln(N_L) + \beta \Delta G \tag{2}$$

and

$$\beta F_{att} = -\ln\left[\sum_{N_\phi=0}^{\min(N_L,N_R)} \binom{N_L}{N_\phi}\binom{N_R}{N_\phi} N_\phi! \exp\left(-N_\phi \beta \Delta G\right)\right], \tag{3}$$

where $N_\phi$ is the number of bonds between the ligands on the nanoparticle and surface receptors, $N_L(N_R)$ is the number of interacting ligands (receptors) and $\Delta G$ the free-energy for the formation of a single bond. Although not crucial for the arguments outlined here, we notice that in both writing Eq. (2) and Eq. (3) we calculate the binding energy for a fixed orientation of the particle, and that $N_L$ and $N_R$ should be interpreted as the number of ligands and receptors, respectively, in the contact region between the nanoparticle and the binding site. In other words, these are the ligands and receptors that, given a certain nanoparticle orientation, can form bonds, and not their total number on the nanoparticle or adsorption site (see details in the Supplementary Notes I and II). Let us now discuss the origin of the two different formulas for the binding energy $F_{att}$. The first form, Eq. (2), is derived under the assumption that at any given time only a single ligand can be bound to a receptor on the surface[17]. This is the case, where a multivalent particle has an interligand distance larger than both its radius and its ligands' average length, a scenario first called by Kitov and Bundle as the indifferent binding scenario[17]. The second expression is instead calculated in the opposite case, the radial binding regime[17], where potentially all $N_L$ ligands can bind all the $N_R$ receptors. However, it should be noted that even in this case two receptors cannot be bound to the same ligand at the same time (and vice versa), i.e. the valence-limited condition of ligand–receptor interactions is correctly preserved[14]. In practice, these two cases represent the minimum and maximum possible gain in multivalent binding and all other possible binding scenarios will provide $F_{att}$ values in between these two.

Having described the attractive contribution in the system, we now turn to consider the second part, the repulsive term arising from excluded volume interactions between the receptor and the polymer brush protecting the nanoparticle. To provide a possible approximation, we use a model first derived in[9], built by combining previous results from Halperin[23] and Zhulina[24,25], to calculate the repulsive free-energy to insert an object in a polymer brush on a curved surface (details of the derivation can be found in the original paper, i.e. ref. [9]). Within this model, we obtain for the contribution of the receptors to the repulsive energy:

$$\beta F_{rep}^{rec} = A(z)N_R \tag{4}$$

$$A(z) = V_R\left[\sigma_0\left(1 + \delta(z)\left[\left(1 + \frac{(\gamma+2)N}{3R_{np}}\left(\frac{\nu a^2}{3\sigma_0}\right)^{\frac{1}{3}}\right)^{\frac{3}{\gamma+2}} - 1\right]\right)^{\gamma-1}\right]^{-\frac{3}{2}}\left(1 - \delta(z)^2\right)^{\frac{9}{4}} \tag{5}$$

where $V_R$ is the volume of the receptor, $\sigma_0$ the average area per polymer chain, $\delta = (z/h_0) \in [0, 1]$ the distance between the nanoparticle and the surface scaled by the average brush height $h_0 = N(\nu a^2/3\sigma_0)^{1/3}$ when grafted on a planar surface, $N$ the degree of polymerisation, $\nu = a^3$ the volume of a monomer of size $a$ and finally $\gamma$ is a parameter that depends on the radius of the nanoparticle core $R_{np}$ with respect to the brush height, and is $\gamma = 3$ for $h_0/R_{np} > (\sqrt{3} - 1)$ and $\gamma = (1 + h_0/R_{np})^2$ otherwise.

For generality, a repulsive contribution from ligands should also be included, for which we would thus have:

$$\beta F_{rep}^{lig} = BN_L. \tag{6}$$

The value of $B$ depends on the exact repulsive mechanism at play and on the specifics of the system. For example, if the grafting surface where receptors reside is also covered by a polymer coating, $B$ would have the same functional form as in Eq. (5) (but with different parameters). We should note, however, that even in the absence of any brush a repulsion should always be expected from excluded volume effects arising from the need to confine the ligands (or receptors) between the surface of the nanoparticle and the grafting surface[14]. For reasons that will be clear later, it is important to highlight that the exact form of $F_{rep}$ as a function of the number of receptors $N_R$ (or ligands, $N_L$) is not crucial to observe range selectivity. In fact, one should expect this phenomenon as long as $F_{rep}$ grows faster than logarithmically, e.g. as a power law as in Eqs. (5), (6). In this regard, it should be reminded that in a mean-field approximation, which should always be valid as long as receptors (ligands) are not too close to each other, the repulsive contribution will always be simply proportional to their number, hence growing much faster than logarithmically.

**Numerical modelling of the influence of various parameters**. In Fig. 2 we show some representative examples of a parametric study on the dependence of the binding probability $\theta$ as a function of the number of receptors (at fixed number of ligands), for the two limiting binding scenarios described by Eq. (2) and Eq. (3), respectively.

Qualitatively, what we see is that the adsorption probability always has a characteristic non-monotonic behaviour. The range of receptor numbers in which adsorption is above a certain threshold value is always finite and can be controlled by tuning both the repulsive and attractive contributions. For example, decreasing the repulsive contribution, e.g. by using receptors of smaller volume (Fig. 2c), or a less dense protective brush (Fig. 2b), leads to a larger adsorption range. This is because a larger number of receptors will be required to make the repulsive contribution overcompensate the attractive contribution from ligand–receptor bond formation. Tuning the attractive contribution instead changes the receptor range in the opposite way. Hence, if the attractive contribution is decreased, for example by reducing the bond strength (increasing $\Delta G$) (Fig. 2a), the range of adsorption

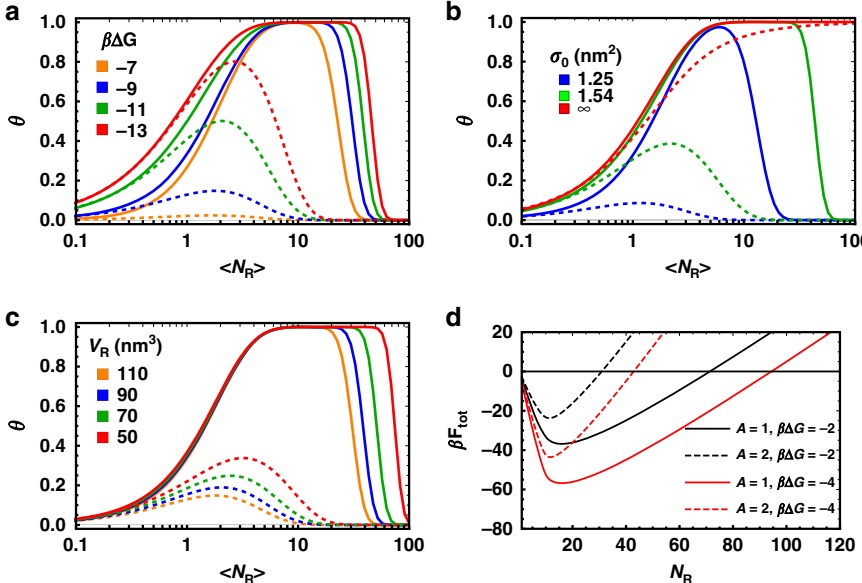

**Fig. 2 Range-selective behaviour as a function of different parameters in the system. a–c** we report the calculated value of the binding probability as a function of the number of receptors for different bond energies $\Delta G$, area per polymer chain, $\sigma_0$ (i.e. the inverse grafting density of the polymer brush) and receptor volume, $V_R$, respectively. Dashed and continuous lines refer to the indifferent Eq. (2) and radial binding scenarios, Eq. (3), which represent the upper and lower bound to the binding free-energy $F_{att}$. Regardless of the binding scenarios the adsorption probability $\theta$ shows a non-monotonic behaviour and binding is only appreciable within a certain range of receptors' number. The various parameters, which are fixed, are chosen to show in each case a range where the non-monotonic behaviour is observed. The nanoparticle size, number of interacting ligands and activity, are kept fixed at $R_{np} = 50$ nm, $N_L = 3$, $z = 10^{-9}$ M, the other values used are $\sigma_0 = 1.95$ nm$^2$, $V_R = 110$ nm$^3$, $\delta = 0.9$; $\beta \Delta G = -9$, $V_R = 40$ nm$^3$, $L = 9$ nm and $\beta \Delta G = -9$, $\sigma_0 = 1.95$ nm$^2$, $\delta = 0.9$ in panel **a**, **b** and **c**, respectively. Note that in panel **b** we keep fixed the value of the length of the ligand rather than the insertion ratio $\delta$. In all cases, we assume a zero contribution to the repulsive energy from the ligands, but its inclusion would only result in a rescaling of the activity $z$ to lower values, from $z \rightarrow z \exp(-F_{rep}^{ligands})$, which does not affect trends observed here. **d** Total adsorption energy $F_{tot}$ for different values of the repulsion parameter $A$ in Eq. (4) and $\Delta G$, assuming a radial binding scenario and using $N_L = 10$. Note that for illustration purposes the scales in panels **a–c** and in panel **d** are different (logarithmic vs linear). As $A$ becomes larger the repulsion increases and as a result the minimum of the adsorption energy decreases (in absolute value) and shifts to lower $N_R$. The opposite trend is observed by increasing the bond strength, which increases attraction. The non-monotonic behaviour of the adsorption energy observed here rationalises the trends observed in **a–c**.

decreases. Curiously, we also notice that for certain combinations of parameters the adsorption probability never saturates to its maximum value. In fact, the peak value of $\theta$ and the range of adsorption are positively correlated and can be tuned in the same way, that is, increasing repulsion leads to lower peak values and increasing attraction leads to higher ones. Noticeably, all these behaviours could be exploited for improving targeting selectivity where a tightly controlled adsorption is required depending on the receptor population.

**Scaling of the attractive interaction and the general physics behind range selectivity.** What is important to notice in Fig. 2 is that on a qualitative level all adsorption curves show exactly the same behaviour: nanoparticles bind appreciably to the surface only when the average number of receptors on an adsorption site varies between a minimum and a maximum value, but not otherwise. For this reason, we dub this phenomenon range selectivity, to distinguish it from the typical adsorption profile where the binding probability monotonically increases with the number of receptors, and quickly saturates to its maximum value of 1 above a certain number of receptors. Whereas the presence of a minimum value of $N_R$ required for observing appreciable adsorption is somewhat intuitive (we need to have at least some receptors to provide a minimum attraction to counteract the loss in translational entropy upon binding), the reason why a growing number of receptors at some point decreases the probability for binding is probably less so, but can be qualitatively understood by looking at how the combinatorial binding entropy of the system

(also called the binding avidity contribution to the free-energy[17]) changes depending on the number of receptors available in the different binding scenarios. Let us first discuss the case of the indifferent binding scenario described by Eq. (2), which provides our lower bound for the attractive contribution $F_{att}$. Because there is only ever one bound ligand regardless of the number of receptors $N_R$, the number of binding configurations scales linearly with this quantity for all values of $N_R$. For this reason the binding entropy, and thus $F_{att}$, grows logarithmically with it. This is in contrast with the repulsive term, $F_{rep}$, which given Eq. (5) grows linearly with $N_R$, because once even a single receptor is bound, all receptors that interact with the nanoparticle will compress the brush, see Fig. 1 for clarity. Hence, $\beta F_{tot} = \beta F_{att} + \beta F_{rep} \approx -\ln(N_R) + A N_R + C$, where $A > 0$ and $C$ are prefactors that depend on the various system parameters, but not $N_R$. Regardless of the values of these prefactors, the crucial thing is that for large enough $N_R$, $F_{tot}$ will always be too high to compensate the loss of translational entropy upon adsorption (as measured by the activity $z$, Eq. (1)) and particles will not bind to the surface anymore, preferring to remain in the bulk solution. The other limiting scenario, radial binding, is more interesting as in this case the growth of $F_{att}$, and hence its influence on $F_{tot}$, shows different regimes depending on the value of $N_R$. Although a precise calculation of $F_{att}$ requires the use of Eq. (3), this expression masks the physics of the problem, which can be more easily captured using the following mean-field arguments (see also Supplementary Notes III). When the number of receptors is much smaller than the number of ligands, $N_R \ll N_L$ receptors bind almost

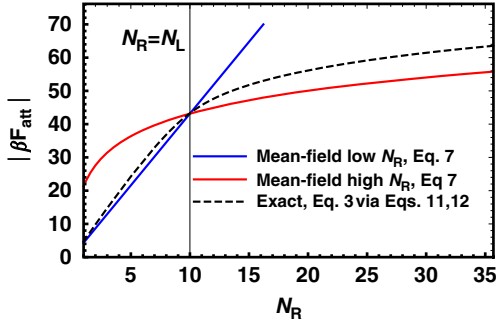

**Fig. 3 Limiting scaling behaviour of the ligand–receptor attractive contribution.** Low ($N_R \ll N_L$, blue) and high ($N_R \gg N_L$, red) limiting behaviours of the binding free-energy $\beta F_{att}$ in the radial binding scenario as calculated via Eq. (3) (using an almost exact approximation via Eqs. (13), (14), see Methods). The black vertical line represents the boundary between the high and low receptor regime, where $N_R = N_L$. Crucially, in the high-receptor regime the free-energy only grows logarithmically with $N_R$, see Eq. (7), unlike the repulsive factor that grows linearly, Eq. (4). For this reason, above a certain receptor number the total free-energy of interaction becomes positive (with respect to nanoparticles in the bulk) and binding is suppressed.

independently from each other because the fact that a neighbouring receptor is bound does not considerably reduce the number of available binding ligands. In this case, the partition function can be factorized to give $q = q_R^{N_R}$, where $q_R$ is the partition function for a single receptor. Note that in the same limit, but considering the point of view of a ligand, this is not the case: if a neighbouring ligand is bound to a receptor and there are few receptors, there will be a high probability that no binding partner is available. Hence, ligands do not bind independently from each other and instead are highly correlated. The symmetric argument holds for the other regime, where $N_R \gg N_L$, and we thus obtain the limiting expression (see also the Supplementary Notes III):

$$\beta F_{att} = \begin{cases} -N_R \ln\left[1 + N_L \exp(-\beta\Delta G)\right], & N_R \ll N_L \\ -N_L \ln\left[1 + N_R \exp(-\beta\Delta G)\right], & N_R \gg N_L \end{cases} \quad (7)$$

which as can be observed in Fig. 3 very well captures the behaviour of $F_{att}$ in their appropriate regimes. What is crucial for the appearance of range selectivity regardless of the binding scenario is that even assuming radial binding, which provides the upper limit for the attractive contribution, we still observe that for large enough $N_R$ the attractive contribution scales only logarithmically. For this reason, at least as long as the repulsive contribution grows faster than logarithmically (as observed in relevant physical models for repulsion), we should always expect that above a critical number of receptors the increase in attraction will be overcompensated by the increase in repulsion. Hence, above this value $F_{tot}$ must start to increase, explaining the origin of the non-monotonic behaviour observed in Fig. 2d) and thus the reduction in the binding probability.

Because our analysis only requires reasonable and physically justifiable assumptions on the scaling of the repulsive and attractive free-energy contributions with the number of receptors but, crucially, does not depend on chemistry-specific details of the system (e.g., the value of the single-bond energy $\Delta G$), we expect range selectivity to be a pretty robust phenomenon observable in various multivalent systems. In our description of the repulsive part, we chose to take the specific case of a polymer coating compressed by receptors because it is a representative example of many applications involving nanoparticles. Although within our model this specific system provides a repulsion linearly scaling with $N_R$, from the previous discussion of $F_{att}$ it is clear that any

form of the repulsive potential growing faster than logarithmically, a broad assumption, would give the same behaviour. It is important to notice that here we assume that the driving force for adhesion is ligand–receptor bond formation but we do not include non-specific interactions between the particle and the surface. For this reason, we expect this phenomena to be relevant in systems where ligand–receptor bond formation is the driving force for binding, which includes a large variety of biological systems, especially where binding is specific.

**Experimental validation using nanoparticles adsorption on cells.** Although we illustrated range selectivity describing the case of polymer-coated nanoparticles binding to a receptor-coated surface, by symmetry such behaviour must also arise in the equivalent scenario where one studies the adsorption probability at fixed receptor numbers but varying the amount of ligands. This is because of the linear term in the repulsive energy as a function of the number of ligands, Eq. (6). Although in this case the repulsion cannot be necessarily attributed to ligand insertion into a brush, we still expect a linear contribution due to the confinement of the ligands in the interacting region between the hard-core of the nanoparticle and the cell surface. This symmetry allows us to validate our theoretical prediction with a more controllable experiment, whose results we report in Fig. 4. In this experiment, as nanoparticles we have prepared different polymersomes[26] functionalised with varying quantities of Angiopep-2 ligands (a small peptide binding to the Low Density Lipoprotein Receptor-Related Protein 1 (LRP-1),[9]) and a reporter dye. The polymersomes as prepared were then incubated with cancer cells (human hypopharyngeal carcinoma cell line FaDu) expressing Angiopep cognate receptors and their adsorption on the cell surface measured as a function of the grafting density of ligands using light microscopy (details in the Methods section). As clearly visible, the observed behaviour shows the expected non-monotonic trend predicted by our theoretical model. Furthermore, it should be noted that despite using two estimates for the particle size distribution, shown as the two sets of theoretical data in Fig. 4 (see details in the Methods Section), the theoretical model still predicts the non-monotonic behaviour observed in experiments. This observation corroborates the fact that range selectivity is a robust phenomenon, not significantly affected by the system polydispersity. In fact, this robustness should be expected, given that the occurrence of range selectivity only depends on very mild assumptions on the scaling of the repulsive interactions.

**Requirements and limitations to observe range selectivity.** In order to better understand the range of applicability of our results, and their possible implementation in other systems, we would like to discuss a few details and limitations of our approach, highlighting in particular those areas where application of the concepts presented here might need extra care. In this regard, we start by pointing out that although we discuss here the problem from the perspective of a multivalent particle binding to a surface, a similar physical picture would more generally apply for the ligand–receptor-mediated binding of two multivalent constructs. This includes binding of a nanoparticle or polymer to a virus, for example, as in the development of antiviral applications[27,28], or binding of a nanoparticle to a cell, like in the experiments we present here. Having said this, not all systems where binding depends on ligand–receptor bond formation obey the exact same physics described here and an extension of our model might be required. In this regard, an important point worth discussing is that of the mobility of ligands and receptors. Here, we limited our modelling to systems where both ligands

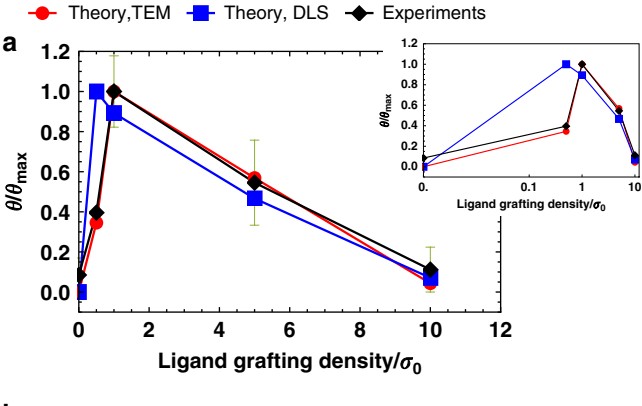

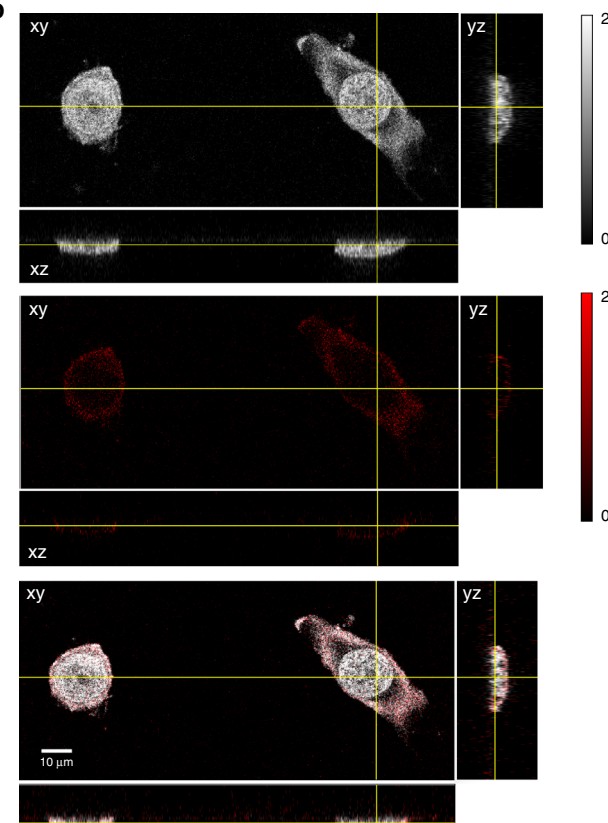

**Fig. 4 Experimental observation of range selectivity and theoretical analysis. a** Adsorption probability $\theta$ (normalised by its maximum value $\theta_{max}$), as a function of grafting density $\sigma$ (normalised by its value at 1% loading $\sigma_O$), comparison of theory vs experimental data. Lines between data points are only a guide to the eye. The theoretical points have been calculated using the expression in Eq. (1), using a Poisson average over both the number of receptors and over the number of ligands, whose average value for the 1% loading of ligands (corresponding to a grafting density of ligands $\sigma_L/\sigma_{ref} = 1$), was used as a fitting parameter. Size polydispersity of the particles was also taken into account, by using the experimentally measured mean and variance at each ligand grafting density, see the Supplementary Methods II. Experimental error bars were calculated as the mean-square root deviation from the average over three independent measurements. See the Method section for more details on both the fitting procedure and the experimental measurements. **b** Orthogonal projections of CellMask™ Green stained cellular membrane after 1 h of incubation with Cy$_5$-polymersomes functionalised with 2% Angiopep ligand. Note how polymersomes only adsorb on the cellular membrane but they do not penetrate in the cell and no fluorescent signal is detectable within the nucleus. For this reason, we can exclude polymersome penetration inside the cell up to the nucleus, at least up to the time of 1 h after incubation with fixed cells, when the confocal microscopy experiments that measure adsorption were run. All confocal microscopy files from which experimental data have been calculated are available in raw form in the Source Data file.

and receptors are fixed. With some prominent exception[29,30], this is true for most synthetic multivalent constructs, including functionalised colloids and nanoparticles. It is also true for certain biological targets. For example, our model would apply to the case where one aims to use ligand-functionalised nanoparticles to target membrane cellular receptors that are cross-linked to the underlying cytoskeleton, and viruses either with fixed receptors or where the receptor density is so high that, although collective moves are possible, the local density of receptor is likely to be approximately fixed[31], e.g. in the coating of the influenza virus. In all these cases, our theoretical framework can be directly applied. When receptor (or ligand) mobility is high and can induce local changes of their grafting density, instead, a theoretical analysis of the binding mechanism should include its effects. In particular, this is also the case for functionalised polymers, where the backbone to which ligands are attached can deform and thus change the local ligand density (albeit with an associated con-figurational entropy penalty). Whereas a full description goes beyond the scope of the present work, we notice that this could be

done using some recent analytical results and computational techniques derived by Mognetti et al. in[32,33]. In practice, we expect range selectivity to still occur but the drop of the binding probability at high densities of receptors (or ligands) to shift to higher values compared to those for the fixed case.

Another relevant question is related to the repulsion required to observe range selectivity. In principle, any force that grows with the number of receptors (or ligands) faster than logarith-mically would be enough. To the best of our knowledge, this includes all known repulsive mechanisms. Practically, however, one needs repulsion to overcome attraction at an experimentally achievable receptor (ligand) density to observe the drop in binding. This can be understood based on the results in Fig. 2d), which show that the smaller the repulsion per receptor (ligand), the larger the range where binding will occur, which could shift the number of required receptors to observe a drop in binding to physically inaccessible values. In this regard, although here we suggest to use a polymer brush because it provides a highly tuneable parameter to control repulsion and observe the non-monotonic behaviour within a specific region, its presence is not a strict requirement. Even in its complete absence, the small ($\approx k_B T$) repulsion due to ligands or receptors being confined within the binding region between the surface of the nanoparticle and that of the adsorption site (see Fig. 1 for reference), which limits their allowed microscopic configurations causing steric repulsion[14], might be enough. For example, this is the mechanism we invoke to justify the decrease in binding with the number of ligands in our experiments.

Regarding the potential role of the brush, it should also be noted that although it allows to control repulsion, burying the ligand too deep inside it could lead to a significant kinetic barrier, affecting the timescale required to observe the equilibrium behaviour we describe. The kinetic barrier in this case stems from the fact that before a receptor can bind to a ligand and recover part of the free-energy through bond formation, the brush must be compressed. In our experiments, a rough order of magnitude estimate (see also Supplementary Notes V) gives a value for this timescale between $10^{-2}$s and $10^{-1}$s, well below the experimental timescale for the adsorption measurements of 1 h.

In general, however, the typical timescale depends on the nanoparticle concentration and on the total repulsive contribution (see the Supplementary Notes V).

In order to implement our results for the development of applications, e.g. for targeted drug delivery, some fine-tuning of the system is also required. For example, for selective drug-targeting one might want to restrict binding to a relatively small range of receptor densities. In the various example presented in Fig. 2, this range varies between about 2 and 3 orders of magnitude, which might be too large. However, we point out that we have made no efforts trying to optimise it. For example, the results in Fig. 2 suggest that the higher the repulsion per receptor (ligand), the smaller the range where binding occurs. More generally, optimisation should be done within the experimental constraints by using all the available parameter space, which could be done with known minimization algorithms see, e.g. ref. [34]. This includes the number of ligands or their binding constant, as well as characteristics of the repulsive brush such as the grafting density or degree of polymerisation ($\sigma_0^{-2}$ and $N$ in Eq.5).

**Range selectivity puts a limit on the optimal grafting density to achieve maximum binding in multivalent constructs**. Also connected to the development of applications, we note that often multivalent constructs have been developed not to target a specific receptor range but rather to simply increase the overall binding strength[35]. This in turn allows to decrease the detection limit of a specific multivalent target, for example, an analyte in solution, useful for diagnostic purposes[36]. Even in this case, our theoretical analysis suggests a general point that could be translated into a design principle: increasing the number of interacting ligands, or in other words the ligand grafting density, beyond a certain value can actually be detrimental to binding, even when the density is not large enough to foresee any potential negative cooperative interaction. Based on our mean-field model we can provide an upper bound to the optimal number of interacting ligands as (see details in the Supplementary Notes VI):

$$N_L^{\text{optimal}} = \frac{N_R}{B} - \chi^{-1}, \qquad (8)$$

where again $B$ is the average repulsion per ligand as defined in Eq. (6) and $\chi = \exp(-\beta \Delta G)$ which, multiplying by the (irrelevant, here) binding volume, is nothing but the binding constant of the given ligand–receptor pair. This formula predicts that for weak enough bonds (larger $\Delta G$), the value of $N_L^{\text{optimal}}$ can become negative. More precisely, this means that, in this case, the binding strength is a monotonically decreasing function of the number of ligands. This is a possibility that is seldom, if ever, considered when discussing the design of multivalent constructs using weak ligand–receptor pairs. It is in fact somehow generally assumed that a higher number of ligands will always yield a higher binding strength. As we show here, this might not be the case as it does not account for the fact that more ligands also bring more repulsion, which might or might not be counterbalanced by the increase in the attractive contribution to the binding free-energy. The same expression also shows that in the other limit, for very strong ligands ($\chi \to \infty$), the optimal number of ligands is still finite (with the caveat that $N_L > N_R$) and depends on the number of receptors as well as the strength of the repulsion, i.e. $N_L^{\text{optimal}} = \frac{N_R}{B}$. This latter prediction, as well as some of the trends presented in Fig. 2, are difficult to test in a systematic way within our experimental system. However, we hope these results will spur further interest towards this goal. In this regard, we suggest that using fully synthetic systems where binding is mediated by ligand–receptor interactions, e.g. DNA-coated colloids and

surfaces, would provide the perfect platform to further test these results in a more controlled manner.

**Range selectivity vs other peculiar types of binding**. We would like now to ultimately discuss our results in relation to other peculiarities of multivalent binding, as well as to draw a distinction between range selectivity as described here and non-monotonic binding reported in literature for protein or antibody/antigen binding[37]. Multivalent constructs can display, under specific conditions, so-called super-selective binding[6,9,10], i.e. a sharp (superlinear) response to receptor density gradients. Here we show that multivalent constructs can also display range selectivity but the two properties are independent from each other (see also Supplementary Notes IV). In fact, range selectivity is somehow a more robust, general phenomenon in the sense that it requires less stringent conditions to be observed. For example, it occurs also for monovalent constructs and does not require weak bonds, as necessary to observe super-selectivity. Thus in general, if the conditions for super-selectivity are met, this phenomenon can be observed together with range selectivity, whereas the opposite is not necessarily true, see also the Supplementary Notes IV. On a different note, we want to point out that we are not the first in describing a decrease in binding at high receptors concentration. This phenomenon, in fact, has long been recognised and discussed in the literature, in particular in solid-phase binding assays for proteins, see, e.g. ref. [37] and references therein. However, the mechanism attributed to non-monotonic binding in these experiments is different from what we describe here. In those cases, it is assumed that at high receptors density the proximity between different receptors leads to limiting the accessibility of their binding sites for the ligands. This is equivalent to making the single-bond strength depend on density, more precisely, decreasing in an anti-cooperative way for increasing densities. For this reason, this phenomenon can only occur for very high grafting densities of receptors and is independent from the properties of the bound construct. In our case, no cooperative effect is invoked ($\Delta G$ is assumed independent of receptor/ligand density) and it is the relative number of ligands vs receptors what matters because it dictates the way the entropy of binding grows in the system. When we account for this, we still conclude that a decrease in adsorption probability must occur due to the different scaling of bond-mediated attraction and repulsion. Importantly, we also show that the mechanism we describe here can be tuned by changing the properties of the binding construct, independently on the targeted surface.

**Concluding remarks and speculations**. In conclusion, we introduce here the concept of range selectivity, whereby a ligand-coated object binds a receptor-functionalised surface only when the latter has a number of receptors within a specific range, but neither below it (which is trivial) nor, more strikingly, above it. Our analysis was based on a combination of a statistical mechanical model for ligand–receptor-mediated interactions[6,13] and a linear form for the repulsive energy, although as extensively explained this phenomenon can be observed for a broad range of choices for this latter contribution. Besides providing a potential explanation for the observation of non-monotonic binding in other multivalent systems presented in the literature[27], we speculate that this mechanism could also arise in the regulation of the interaction between biologically entities, which often interact exploiting multivalency. For example, cells could counter-intuitively enforce the unbinding of a multivalent construct from their surface, e.g. a drug-carrying nanoparticle or another cell, by increasing the number of receptors usually employed by this construct for binding. Curiously, this mechanism could be

potentially exploited by cancer cells to avoid being recognised and attacked, especially considering that they usually over-express receptors, which will thus be present at high density on their surface. More generally, we suggest to consider the potential of range selectivity when designing multivalency-based applications, in particular for drug delivery and biosensing applications where it could help to avoid side effects due to off-target binding.

## Methods

**Theoretical model**. Assuming that the attractive contribution $F_{att}$ to the free-energy of adsorption is dominated by bond formation between ligands and receptors, we can write:

$$\beta F_{att} = -\ln \sum_{N_\phi} \Omega(N_\phi) \exp(-N_\phi \beta \Delta G), \qquad (9)$$

where the sum is over all possible number of bonds $N_\phi$. $\Delta G$ is the energy of a single bond (which depends on the specific ligand–receptor pair chosen) and $\Omega(N_\phi)$ is the number of configurations with that specific number of bonds. In order to calculate this quantity, a specific binding scenario must be chosen. Different binding scenarios differ by the number of allowed configurations $\Omega(N_\phi)$ (which measures the avidity entropy[17] via $S = k_B \ln \Omega$). The weakest possible binding contribution is for indifferent binding (see ref. [17] for reference). In this case, only a single ligand can be bound to a receptor at any one time and we have $\Omega(N_\phi) = N_R N_L$ and $N_\phi = 1$, leading to $\beta F_{att} = -\ln(N_R) - \ln(N_L) + \beta \Delta G$, i.e. Eq. (2) in the main text. For the case of radial binding instead all ligands can bind to all receptors (but in each configuration only a single ligand can be bound to any specific receptor, and vice versa), leading to:

$$\Omega(N_\phi) = \binom{N_L}{N_\phi}\binom{N_R}{N_\phi} N_\phi! \qquad (10)$$

and the sum in Eq. (9) extends from 0 to $\min(N_L, N_R)$. This partition function cannot be written in close form and it is not computationally efficient to calculate it with brute force. However, as shown in ref. [13], the ligand–receptor-mediated energy in any possible binding scenario (except for the case where the number of both ligands and receptors are 1 at the same time, ref. [15]), thus including the radial case, can be approximated to within a fraction of $k_B T$ accuracy by the set of coupled equations:

$$\beta F_{att} = \sum_i N_i \left(\ln p_i + \frac{1-p_i}{2}\right) \qquad (11)$$

$$p_i + \sum_j p_i p_j \chi = 1, \qquad (12)$$

where $\chi = \exp(-\beta \Delta G)$ can be interpreted as the single-bond strength[7], which increases for lower values of $\Delta G$. In Eq. (12), the index $i$ refers to any ligand or receptor in the system and the sum is extended overall binding partners $j$ of $i$. Hence, there are $N_L + N_R$ coupled equations to solve. In the radial binding scenario one has that each ligand or receptor has the same number of neighbours (either $N_R$ for ligands or $N_L$ for receptors) and thus the previous equations reduce to two coupled equations only[9]:

$$\begin{cases} p_L + N_R p_L p_R \chi = 1 \\ p_R + N_L p_L p_R \chi = 1 \end{cases}$$

whose simultaneous solution leads to

$$F_{att} = \sum_{i=L,R} N_i \left(\ln p_i + \frac{1-p_i}{2}\right) \qquad (13)$$

$$p_L = \frac{(N_L - N_R)\chi - 1 + \sqrt{4 N_L \chi + (1 + (N_R - N_L)\chi)^2}}{2 N_L \chi} \qquad (14)$$

(and a symmetric formula changing the pedices $L$, $R$ for $p_R$), which we use to plot the curves in Figs. 2– 4 whenever we use the radial binding scenario.

The theoretical points in Fig. 4 have been obtained by fitting the experimental data using the expression in Eqs. (1)–(6), where in this case we have taken a double Poisson average over both the number of receptors per site as well as over the number of interacting ligands, to take into account inhomogeneities in the functionalisation of the polymersomes. For calculating the attractive contribution, we assumed the radial binding scenario. We approximate the binding distance $L$ to be $R_g$, the gyration radius of the ligand treated as a Gaussian chain. This information is also used to estimate the interacting area, which further depends on the particle size, allowing us to introduce the effects of polydispersity in the estimation of the adsorption probability (see the Supplementary Methods II). Furthermore, we have $R = R_{np} + h$, $h = 8$ nm, being the height of the brush, as estimated from the degree of polymerisation of the protective PEG coating and its grafting density using the Zhulina model[24] and $R_{np}$ being the size of the

**Table 1 Nanoparticles size characterisation.**

| Ligand load (%) | $\bar{R}^{TEM}$ (nm) | $\sigma^{TEM}$ (nm) | $PDI^{TEM}$ | $\bar{R}^{DLS}$ (nm) | $\sigma^{DLS}$ (nm) | $PDI^{DLS}$ |
|---|---|---|---|---|---|---|
| 0 | 26.6 | 10.9 | 0.17 | 49.2 | 12.8 | 0.26 |
| 0.5 | 10.0 | 4.7 | 0.21 | 45.6 | 12.3 | 0.27 |
| 1.0 | 11.5 | 5.8 | 0.26 | 43.7 | 12.5 | 0.28 |
| 5.0 | 19.3 | 10.1 | 0.27 | 43.0 | 12.4 | 0.29 |
| 10.0 | 20.4 | 8.6 | 0.18 | 43.4 | 12.7 | 0.29 |

Average polymersome radius ($\bar{R}$), mean-square root deviation ($\sigma$) and corresponding polydispersity index PDI (details in the text) for our samples at different ligand load (in %). Superscripts refer to measurement on the same batches made via TEM and DLS, respectively. As it can be observed, there is approximately a factor of 2 of difference, possibly due to shrinking upon drying for the TEM analysis, see also the discussion in the main text.

nanoparticle as experimentally determined for the different ligand loadings via TEM and DLS, see Table 1 as well as details in the Supplementary Methods II. These numbers also give $\delta = 0.375$ in Eq. (5) and $\gamma = (1 + h_0/R)$ for the estimation of the repulsive contribution due to receptors via Eq. (5), for which we further used $V_R = 188$ nm$^3$ for the Angiopep receptor, as estimated from known structural data[9]. Considering that the ligand repulsion should be due to its interaction with the impenetrable surface of the cell, an estimate for $B$ can be obtained by assuming the ligand behaves as a Gaussian chain at a distance $R_g$ from a flat plane, using the formulas and parameters reported in ref. [38], which give: $B(r) = a \exp(-b(r/R_g - c))$, $r$ being the distance from the plane. Using $a = 3.1995$, $b = 4.1662$, $c = 0.4996$ (these parameters were fit in ref. [38] to reproduce exact Monte Carlo data for repulsion up to $10 k_B T$), for $r = R_g$ we obtain $B = 0.40$. This leaves two fitting parameters: the reference grafting density of ligands on the surface for the polymersomes prepared at 1% loading of ligands $\sigma_{L,ref}$ (since we only know the ratio between different polymersomes at different ligand loading, but not their absolute value) and the average grafting density of receptors $\sigma_R$. Given these formulas, we have fitted the experimental data using a Monte Carlo annealing to minimise the quantity $E = \sum_i w_i (\theta'_{i,exp} - \theta'_{i,theory})^2 / \sum_i w_i$, where for $w_i$ we take the inverse of the m.s.r.d of each experimental data $\theta_{i,exp}$, the experimentally measured adsorption normalised by its maximum value among all polymersomes of different grafting densities. The procedure started with an effective temperature of 1, scaling the temperature by a factor of 0.95 every 100 MC sweeps until the temperature reaches a value of $10^{-7}$, at which point the system has already ceased to evolve.

**Result of fitting the experimental data**. The overall procedure outlined above produced fitting parameters of $\sigma_L = 2.0 10^{-2}$/nm$^2$ and $\sigma_L = 1.9 10^{-2}$/nm$^2$ using the size distribution from TEM and DLS, respectively, equivalent to between [0–24] ligands at the grafting densities considered. The grafting density was estimated by fitting the number of interacting ligands and then assuming that all ligands grafted within a distance of $2R_g$ from the binding site where available for binding and that the equilibrium distance between the surface of the nanoparticle and that of the binding site was $R_g$, which thus provides an estimate of the interacting area on the polymersome of $A_{int}^{np} = 2\pi R_{np} R_g$ [39] (see Fig. 6 in the Supplementary Notes I for clarity). The fitted density of receptors was $\rho_R = 1.5 10^{-4}$/nm$^2$ and $\rho_R = 2.1 10^{-4}$/nm$^2$ using the size distribution from TEM and DLS, respectively, equivalent to ~$10^{-2}$ receptors per interacting area on the adsorption site depending on the polymersome size, estimated as the projection of $A_{int}^{np}$ on the flat adsorption surface and thus equal to $A_{int}^{surf} = \pi \left[R_{np}^2 - (R_{np} - R_g)^2\right]$. For what concerns the grafting density of ligands on the polymersomes and receptors on the cell membrane, the fitted values we obtain are within an order of magnitude, and thus consistent with our coarse-grained description, previous independent estimates obtained for the same system of $\approx 1.3 10^{-3}$/nm$^2$ and $2.1 10^{-5}$/nm$^2$, see[9]. Finally, it should be noted how TEM and DLS data, despite giving an estimate of the size of the polymersomes differing by about a factor of 2, still provide consistent estimates for both $\sigma_L$ and $\sigma_R$, showing how the theoretical model and its results are not significantly affected by details of the particles size distribution.

**Preparation of polymersomes**. Poly(ethylene glycol)-block-poly(2-(diisopropyl amino) ethyl methacrylate) (PEG-b-PDPA) and $N_3$-PEG-b-PDPA copolymers were synthesised as previously reported by the atom-transfer radical polymerisation method[40,41]. For the fluorescent-labelling ($Cy_5$-PEG$_{113}$-PDPA$_{100}$) and ligand-conjugation (Angiopep$_2$-PEG$_{68}$-PDPA$_{90}$) one eq of $N_3$-PEG-b-PDPA was first assembled in phosphate-buffered saline (PBS) by pH switch procedure[42]. The solution of self-assembled polymer was then degassed by sonication and inert gas flow under stirring. The degassed solution was mixed with 1.2 eq of the corresponding ligand. For peptide conjugation (AP-alkyne), the peptide was dissolved in degassed PBS pH 7.4, whereas water insoluble ligands such as $Cy_5$-alkyne were added in degassed dimethylsulfoxide (DMSO) having a final DMSO:PBS ratio of

10:1. Then sodium ascorbate (5 eq) was added and the mixture was further degassed for at least 30 min. Finally, 1 eq of CuSO₄ was added under inert atmosphere and the reaction was left reacting at 40 °C for 72 h protected from light. Dialysis of the labelled polymers was done against DMSO and then water to purify them (MWCO at least 5kDa for peptide purification and 3.5 kDa for dye purification). The labelled polymers were recovered after lyophilisation. For polymersome preparation with increasing amount of ligand, the co-polymer $PEG_{113}$-$PDPA_{80}$ was mixed with $Angiopep_2$-$PEG_{68}$-$PDPA_{90}$ (0–10 mol%) and $Cy_5$-$PEG_{113}$-$PDPA_{100}$ (10 mol%) and the mixtures were dissolved in tetrahydrofuran/dimethylsulfoxide (90:10) at a final total polymer concentration of 20 mg/mL. 2.3 mL of PBS pH 7.4 (aqueous phase) were pumped at 2 μL/min into each organic solution using an automated syringe pump. The addition of the aqueous phase was carried out under continuous stirring at 40 °C. After the injection, an additional extra volume of PBS (pH 7.4) (3.7 mL) was added manually. In order to remove the remaining organic solvent, the polymersome dispersions were transferred in a cellulose semipermeable membrane (3.5 kDa cut-off) and dialysed in PBS (pH 7.4) for over 24 hours at room temperature. The samples were centrifuged at 1000 r.c.f for 10 min, sonicated at 4 °C for 20 min and purified through a size-exclusion chromatography (SEC) column packed with Sepharose 4B. All the samples were stored at 4 °C and protected from light until further use. All PEO (o PEG) materials were purchased from Iris Biotech. All solvents were ordered from Sigma–Aldrich and used directly as provided unless specified. Copper sulfate and sodium ascorbate were obtained from Sigma–Aldrich. $Cy_5$-alkyne was ordered from Lumiprobe. Angiopepalkyne (Propargyl-TFFYGGSRGKRNNFKTEEY) was purchased from Genscript. Dynamic Light Scattering (DLS) was used to confirm the colloidal stability of all the preparations as well as to provide a measurement of the hydrodynamic radius of our particles, that we can use as an upper bound to their real size (see Supplementary Figs. I and II in the Supplementary Methods II). DLS measurements were performed using a Malvern Zetasizer equipped with a He–Ne 4 mW 633 nm laser, diluting the polymersomes solution with PBS (pH 7.4) in disposable polystyrene cuvettes. Transmission Electron Microscopy (TEM) was used to obtain the bare particle size, which due to potential shrinking upon drying has been used as a lower bound to the actual particle size in the calculations, to be compared to the upper limit provided by the DLS data. Their size distribution as measured by TEM and an estimate of the polydispersity index, $PDI = (\mu/\sigma)^2$ ($\mu$ and $\sigma$ being the average and variance of the size distribution) were extrapolated with a custom-made algorithm implemented on Matlab® for image analysis. For TEM analysis, polymersomes were deposited for 1 min on glow-discharged carbon-coated copper grids and then stained with a PTA solution at 0.5 % (w/v) for 2 s. Details of the DLS and TEM analysis, including the Matlab algorithm used to analyse the latter, can be found in the Supplementary Methods II and III, whereas here we only report the values of the average polymersome size and the PDI obtained in Table 1. Clearly, such values indicate a relatively high degree of polydispersity in our system. It should also be noted that, in the TEM data, our samples present a relatively large variation in the mean radius depending on the ligand loading, which varies between around 27 nm and 10 nm. Note that this variability in size distribution is also taken into account in the fitting to the experimental data.

**Measurement of adsorption probability**. FaDu cells (ATCC HTB-43) were seeded on an eight-well chamber slide (iBidi) at a density of 20,000 cells per well and maintained in MEME (Minimum Essential Medium Eagle M5650-Sigma) supplemented with 10% Fetal Bovine Serum (Sigma–Aldrich) and 1% penicillin/streptomycin (Sigma–Aldrich) at 37 °C in 5% $CO_2$. After 24 h, the media was removed, cells were washed three times with Dulbecco's Phosphate-Buffered Saline (DPBS) and fixed with 3.7% (V/V) of paraformaldehyde (v/v in DPBS) for 10 min at room temperature before incubation with polymersomes. The fixing process with paraformaldehyde (PFA) cross-links molecules by forming covalent chemical bonds between proteins and creating an insoluble mesh that preserves cellular architecture and composition, including the presence of receptors on the plasma membrane, and it also prevents any process of endocytosis. For this reason, in analysing our data we only took into account the binding of the polymersome on the cellular surface but not their internalisation. Fixed cells were incubated for 1 h at 37 °C with $Cy_5$-labelled and $Angiopep_2$-decorated polymersomes (0.15 mg/mL) in PBS (pH 7.4). After 1 h, polymersome dispersions were removed, cells were washed three times with DPBS and treated for 5 min at room temperature with CellMask Green (1:1000 in DPBS). Cells were left in Live Imaging Solution and the adsorption of polymersomes on cell membranes was analysed on a Leica SP8 confocal laser scanning microscope with ×63 oil immersion lens. The total emission fluorescence of $Cy_5$-labelled polymersomes was measured in the 650–700 nm range using an excitation wavelength of 633 nm. Specifically, in every single test, for each formulation (corresponding to a given Angiopep2 percentage), the polymersome fluorescence was collected by scanning the whole thickness of the cell using a z stack of 30 images. Every stack could include more than one cell and the fluorescence measurement per formulation was carried out on a minimum number of 40 cells. Cells that were fixed during mitotic events were discarded from the analysis. The total polymersome fluorescence of each stack was normalised by the number of cells and these normalised values were averaged. All the experiments were done in triplicates. In order to make the image analysis automated a custom-made script implemented on Matlab® was used.

## Data availability

Source data used for producing the figures in this manuscript and in the Supplementary Information are provided with this paper. Due to their large size (TBytes), confocal microscopy images have been stored on a local server and are available from the authors upon reasonable request. Source data are provided with this paper.

## Code availability

The computer code used to generate the graphs in this manuscript and in the Supplementary Information is available in the Source Data file provided with this paper.

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

## Acknowledgements

S.A.-U. would like to acknowledge the Chinese Academy of Science for a visiting scientist fellowship via the PIFI program. L.R.-A. thanks the ESPRC for the funding of her salary (Project EP/N026322/1). G.B. thanks the EPSRC (EP/N026322/1) for funding part of his salary via an Established Career Fellowship and the ERC (CheSSTag 769798) for his consolidator award, and the Children with Cancer UK (Grant number 16-227). S.A.-U. and M.L. also thank the Beijing University of Chemical Technology CHEM-CLOUDCOMPUTING Platform for computational support.

## Author contributions

M.L. performed the numerical calculations and analysed the data; A.A. and L.R.-A. optimized and carried out the cell experiments, performed the microscopy measurements and analysed the data; G.M. analysed the TEM data; M.S. prepared the polymersomes; A.P. synthesized the polymers; E.S. and L.R.-A. supervised the experimental part and the polymersome preparation. M.S. and L.R.-A. characterised the polymersomes. G.B. conceived and supervised the experimental part and analysed the data. S.A.-U. conceived the original idea, developed the theoretical model, analysed numerical calculations and the experimental data. M.L., A.A., L.R.-A., G.B. and S.A.-U. revised the manuscript. M.L., G.B. and S.A.-U. wrote the paper.

## Competing interests

The authors declare no competing interests.
