## [Peer Review File · Nature Communications]

Reviewers' comments:

Reviewer #1 (Remarks to the Author):

Liu et al. have investigated the ligand-receptor binding properties arise from the multivalent nature of interactions. Specifically, the peculiarity of binding nanoparticles only to targets carrying a specified range of receptors is described using statistical mechanical modeling and experiments, which they called as range selectivity. The authors consider the adsorption of multivalent nanoparticles on a receptor decorated surface. The nanoparticles are covered with ligands complementary to the receptor and protective polymer brush. The article focuses on how one can control the adsorption of nanoparticles covered with ligands complementary to surface receptors.

The authors come up with a nice idea where they consider the interplay between attractive and repulsive energy by tuning the density of grafting polymer brush, receptor volume, bond energy, however they did not illustrate the geometric details such as the size distribution, maximum contact area on the surface if the surface is undeformed. The work requires further systematic simulation and/or experimental data to prove the theory. The authors may look at the article where detail experimental and corresponding theory was given earlier by Parveen N. et al. *Langmuir* 33, 4049 (2017), and *J. Am. Chem. Soc.* 2019 using multivalent viruses. I believe that this work requires further simulation and/or experiments to be published in a journal with a broad readership. Addressing the following question may improve the quality of the article.

Q. Authors report the size of the nanoparticle determined experimentally. Polydispersity in the size of the particles may exist and may impact the results. It is recommended to show the distribution data in SI, could you please comment.

Q. The author claims that they performed a confocal laser scanning microscope for imaging. It is recommended to show some of the imaging data and corresponding analysis to emphasize the experimental findings and for in general, improve the article for readership.

Q. In Figure 2d, the energy vs. number of receptors plot, the number of receptors reaches up to 100. Even for repulsive interaction, polymer brushes (with or without ligand sites) need to make contact with the receptors on the surface. A particle of the given size and grafting density, please calculate how many receptors will be in contact with the polymer chains — comment whether the surface is flat or bend, if so, how that may affect this number.

Q. From the fitting of experimental data, it is found that polymer brushes are larger (8 nm) than the tether, which binds the ligand to the nanoparticle (6 nm). Therefore, I would expect an early repulsion between the polymer brush and the receptor before there is the formation of a ligand-receptor bond. Please comment on that.

Q. On page 5, 'NL should be interpreted as'. It must be written that NR is also the number of receptors present in the contact region between the nanoparticle and surface.

Q. In equation (3), N_{ϕ} is the number of bonds between nanoparticle and surface receptors, which should be mentioned immediately after the equation. I assume that the authors consider the upper limit of $N_{\phi} = \min(NL, NR)$ instead of $\max(NL, NR)$.

Q. In Figure 2a, b, c author should mention the value of the parameters which are kept fixed, for example, in Figure 2a, the value of A used in the calculation.

In Figure 2d, changing the Δ_G , at a fixed A, there is an insignificant change in the NR at which the minima of F_{tot} appear. But in Figure 2a, why do we see the different saturation NR as a function of Δ_G ? Could you repeat the plot of Figure 2d for the same parameter space used in Figure 2a? Mention the unit of A and Δ_G in Figure 2d.

I assume in Figure 2, the distance between nanoparticle and surface remains fixed. How do the authors choose the distance?

Q. In equations 9 and 10, what is p? I assume that pi is the fraction of free ligand or receptor at a particular orientation. It means even at a certain number of NR, there is a repulsive term due to free linkers or receptors? How do you incorporate it into the model?

minor comments—

1. Page 14, Last line " in Fig24..." should be "in Fig2 and Fig4 .."

2. If you use VR as the volume of the receptor, please use the same notation throughout the article. Same for other parameters.

Reviewer #2 (Remarks to the Author):

The authors propose a generic mechanism to generate what they call 'range selectivity' in the recognition of receptor densities on (cell) surfaces by multivalent nanoparticles (NPs). The mechanism is based on two opposing forces: attraction derives from ligands on the NP interacting with a few receptors on the surface, whereas repulsion derives from a repulsive brush (or any other repulsive medium) on the NP interacting with all receptors within the interaction range of the NP. The authors argue that the balance of attractive and repulsive interactions leads to a non-monotonous dependence of probe binding on receptor surface coverage with a maximum at intermediate coverage and gradually reduced binding towards lower and higher coverages.

The idea certainly has its merit, and is potentially of interest to a broad audience. However, I have a number of concerns with this study which I am afraid will require extensive revision to address. Some of the points of critique listed below refer to the quality and comprehensiveness of the presented data, and others to the practical usefulness of the proposed approach as well as style:

- The experimental data provided are very limited, making it impossible to assess the rigour of the experimental work and data analysis, and the relevance of the theoretical model for the experimental system. Amongst the questions that remain open are:

(a) How did the authors confirm the ligand labelled polymers (i) mix well with the non-labelled and fluorescently labelled polymers and (ii) do not affect polymersome size? Both aspects are important to ascertain homogeneous dilution of the ligands within the polymersomes and that the polymer mixing ratio is indeed a reliable proxy of relative ligand density as the authors assume.

(b) How monodisperse were the polymersomes, and have the authors considered how size-preferences in binding could bias the results?

(c) How did the nanoparticles distribute on the cell surfaces? Did the authors observe nanoparticle uptake in addition to binding, and if so, how did the authors discriminate between these two scenarios? Representative fluorescence micrographs of the polymersomes on cells need to be shown, and adequate analysis of the data demonstrated.

(d) I find the symmetry argument on page 11 (second paragraph) confusing. In order for the experimental model (in which the ligand density is varied) to be 'symmetric' to the theoretical model (in which the receptor density is varied), I would have expected that the repulsive brush should also switch location (from the nanoparticle in the theoretical model to the cell surface in the experiment). The free energies of insertion into the polymer brush would likely be quite different if the objects to insert and the polymers forming the brush were tethered to the same or different surfaces?

(e) Related to the above it may also be worth providing the reader with information about the likely location of the ligand within the polymer brush as this should quite sensitively depend on the length of the tether to which the ligand is linked as compared to the length of the polymers forming the brush. Ideally, I would like to see a control experiment with ligands on a long tether and other polymers forming a short brush: in this scenario the reduction in binding at high ligand densities should disappear, or at least be much less pronounced!

(f) Some critical analysis of the results of the fit would be worthwhile. Did the authors attempt to directly measure (or estimate) the receptor surface density on the cells to check if the derived number for ρR is reasonable. Likewise, is the size and organisation of the polymersome consistent with a scenario of 7 interacting ligands within the polymersome area?

(g) Did the authors test if receptor mobility affects nanoparticle binding? This question would appear important, since the authors state in their manuscript (page 11) that lateral mobility should be suppressed for their model to work.

Whilst not all of the above-requested data may fit into the main manuscript it can readily be provided as supplementary information.

- Furthermore, it would be desirable to see theoretical data displayed in a more consistent and complete manner:

(a) The shape of the curves in Figures 2 and 4, for example, looks quite different. Is this entirely due to the different x axes scaling (i.e. log vs linear) or are there further differences that are difficult to appreciate with the heterogeneous axis scaling?

(b) The authors highlight superselectivity in their introduction; a discussion would be worthwhile on how the range selectivity proposed here relates to superselectivity and/or if the two can co-exist. For example, superselectivity is conveniently defined by the parameter $\alpha = d \log \theta / d \log \langle NR \rangle$

exceeding 1; can a similar analysis be performed for range selectivity? To appreciate the co-existence of range selectivity and superselectivity it would be practical to see the data in Fig. 2a-c in log-log plots.

(c) Could Fig. 2b include an example of very high σ_0 to illustrate the critical importance of the repulsive brush for the reduced binding in the limit of at high $\langle NR \rangle$?

(d) Should the label on the x axes in Fig. 2 be $\langle NR \rangle$ throughout?

(e) What were NL, Rnp and L for the example curves shown in Fig. 2?

- 'Range selectivity' appears entirely justified as a term for the reported phenomenon. I notice, however, that the effective binding ranges are generally quite large, typically spanning more than one order of magnitude in receptor density (an exception is the solid blue line in Fig. 2b). The authors may want to comment on this as it would appear quite important a consideration from the perspective of practical utility.

- The title does not appear consistent with the work shown. Firstly, the phenomenon described does not appear to require multivalent constructs, as shown for the 'indifferent binding' scenario which effectively refers to monovalent constructs. Secondly, whilst the repulsive interaction may be of some entropic nature (as for a polymer brush) one could equally well envisage some non-entropic force to play the repulsive part in the 'game'.

- The authors may want to check the literature for examples of experimental systems in which non-monotonous binding of soluble analytes as a function of surface-binding sites is observed. Steric hindrance at high densities of surface binding sites leading to reduced analyte binding has previously been recognised in solid-phase binding assays even for simple analytes such as (monovalent as well as multivalent) proteins. Although in this context the phenomenon is typically considered a nuisance rather than a benefit, it bears the signature of range selectivity.

Reviewer #3 (Remarks to the Author):

Meng Liu and co-authors present a manuscript demonstrating range-selective binding in multivalent binding scenarios. They derive theoretically why, for a very broad class of systems (some requirements are specified but they are very mild and most systems would satisfy them) it should be expected that binding is most prevalent at an intermediate range of receptor densities, and not below (obviously) or above (much less obvious). This would have been publishable by itself. Furthermore, the authors add data from an experimental system involving polymersomes and cancer cells that demonstrates the predicted behavior. These are very important findings that will interest a broad readership from medical biologists to physicists. I therefore recommend this manuscript for publication. The main result is theoretical and does not involve any complex calculation methods, so reproductibility is not an issue (I cannot assess how hard it would be to reproduce the experimental part). Of course, I do have some recommendations that I would like the authors to consider before making the recommendation final.

* It is not clear how equations 4 and 5 are derived. I understand technical details are not desirable in the main text of a letter of this type, but perhaps a few words could be added, or a methods section similar to section A.

* Methods section A refers to figure 24, which does not exist (should be 3).

* A restriction on the range of applicability of the results is detailed on page 11 (about the receptors having to be either fixed in space or laterally immobile). I feel this comes a bit early (it feels as if it should be part of the discussion, but the paragraph after it still contains results), and therefore does not get the attention it needs. Perhaps the authors can add a statement on how generally this requirement is satisfied, and name a few systems for which it might not be? I would also reconsider the order of presentation but I leave this optional because the letter reads quite nicely as it is.

* Equation 7 confuses me: I would think one should either sum over all values of N_{ϕ} and include the Ω , or sum over all configurations ϕ and leave out the Ω . The main text chooses the former. This equation seems to have them mixed up. Please clarify the notation and (if you agree that it's currently not consistent) correct the equation.

* The legends in figure 2 are very small and hard to read.

To all Reviewers,

First of all, we would like to thank you for the constructive comments we have received and for the questions raised. They clearly allowed us to clarify a few points, improve the theoretical analysis and add some context to our results, thus increasing the strength and clarity of our conclusions. In this regard, Referee 1 and 2 in particular ask to see more experimental data for scrutiny. We now provide illustrative examples in the main text and a more in-depth discussion in the new Supplementary Information, which now includes confocal microscopy, TEM and DLS data. We also include *almost* all raw data in an attached DataFile file to allow others full scrutiny of our results.

In the following, we give a point by point answer to your questions, as well as highlighting how we changed the main text in response to them (**these are highlighted in red**). Please note in this regard that **reference numbers used here for citing the literature refer to the bibliography given below** at the end of this response letter, not their number in the main manuscript. We also note that we now have a completely new (we did not have one before) Supplementary Information, and details of the mathematical derivations as well as analysis of the experimental data can be found there. Finally, we also note that we have added one co-author, Gabriele Marchello, who helped us in automatising the analysis of the TEM images reported here.

We look forward to hearing from you again and remain available, will any further clarification be required on our side.

Best,

Stefano Angioletti-Uberti (on behalf of all authors)

Reviewer 1 (Remarks to the Author):

R: Liu et al. have investigated the ligand-receptor binding properties arise from the multivalent nature of interactions. Specifically, the peculiarity of binding nanoparticles only to targets carrying a specified range of receptors is described using statistical mechanical modeling and experiments, which they called as range selectivity. The authors consider the adsorption of multivalent nanoparticles on a receptor decorated surface. The nanoparticles are covered with ligands complementary to the receptor and protective polymer brush. The article focuses on how one can control the adsorption of nanoparticles covered with ligands complementary to surface receptors. The authors come up with a nice idea where they consider the interplay between attractive and repulsive energy by tuning the density of grafting polymer brush, receptor volume, bond energy, however they did not illustrate the geometric details such as the size distribution, maximum contact area on the surface if the surface is undeformed. The work requires further systematic simulation and/or experimental data to prove the theory. The authors may look at the article where detail experimental and corresponding theory was given earlier by Parveen N. et al. *Langmuir* **33**, 4049 (2017), and *J. Am. Chem. Soc.* **2019** using multivalent viruses. I believe that this work requires further simulation and/or experiments to be published in a journal with a broad readership. Addressing the following question may improve the quality of the article.

We thank the reviewer for his/her appreciation of our idea and for the constructive comments. In particular, he/she (similarly to Reviewer 2 and 3) asks for more details describing the geometry of binding (which turns out to give the same interacting area as that calculated in the suggested article, now cited), and the characterisation of the particles used in the experiments, which we now provide. Please see the revised Figures in the main manuscript and the Supporting Information, as well as our answers below.

The maximum contact area depends on the behaviour of the tether used to graft the ligand on the surface and its polymer properties. In our calculations to fit the data of Fig.4, we estimated this contact area by assuming that the tether of the ligand, which sets the distance

between the particle and the surface, behaves as a Gaussian chain with the experimentally measured gyration radius R_g , and that all receptors within $d = 2R_g$ can interact (see the Figure in the new section “Assumed Binding Geometry for fitting” in the Supplementary Information). As in [Parveen N. et al. *Langmuir* 33, 4049 (2017), and *J. Am. Chem. Soc.* 2019], which uses results from [Zhdanov, V. P., and Höök, F., *Eur. Biophys. J.* 44, 219-226 (2015)], this leads to an interacting area of $A_{\text{int}}^{\text{poly}} = 2\pi R_{\text{np}} R_g$ (where R_{np} is the radius of the polymersome) for the polymersome, i.e., the nanoparticle, and an area of $A_{\text{int}}^{\text{site}} = \pi [R_{\text{np}}^2 - (R_{\text{np}} - R_g)^2]$ for the adsorption site.

In any case, it is important to highlight that these choices are only used to obtain a rough, order of magnitude, estimate for the fitting parameters but in no way affect the overall non-monotonic shape of the binding curve. In other words, the occurrence of range selectivity does not depend on these details but the estimated range can slightly differ. Crucially, we never claimed our simple theoretical model to be able to predict quantitatively the binding curve but only the fact that it very robustly predicts the occurrence of range selectivity (and its trends as a function of the system’s parameters), since the non-monotonic binding profile only depends on the scaling behaviour of the different interactions. Quantitative details of the binding curve are instead system-dependent and require detailed molecular models and appropriate computer simulations. However, it is not our intention to show any system-dependent behaviour but rather to present a general and robust mechanism, as also highlighted by Referee 3, and provide a rough estimate of the quantities involved in fitting Fig.4 of the main manuscript. Note that the latter are now also critically analysed in the new version of the “Fitting of experimental data” subsection of the Methods section).

Referee 1 also requires us to answer an extensive set of questions to improve the quality and clarity of the paper and clarify the robustness of our results, as we do below.

R: Authors report the size of the nanoparticle determined experimentally. Polydispersity in the size of the particles may exist and may impact the results. It is recommended to show the distribution data in SI, could you please comment.

As suggested by the referee, in the Supplementary Information of the revised manuscript we now provide an extensive description of the characterisation of the particles used in our

experiments, in particular their size polydispersity as measured via TEM. The data is also now made available in raw form in the DataFile.zip file available in the new submission.

Whereas full details can be seen in the new Supplementary Information in the section “Colloidal stability of the polymersomes, polydispersity and its effect on range selectivity”, we quote here for quick reference that the polydispersity index (PDI), taken here to be given by $PDI = (\bar{\mu}/\bar{\sigma})^2$, $\bar{\mu}$ and $\bar{\sigma}$ being the average size and the square root of its variance, is around 0.25 in our samples, suggesting a relatively high polydispersity in our system.

Polydispersity affects the quantitative value of the binding probability but not the overall shape of the non-monotonic adsorption curve, i.e. the occurrence of range selectivity. Nevertheless, we have performed again the fitting of the experimental data accounting for polydispersity effects, see the new Figure 4 in the main manuscript. In the new calculations, polydispersity affects the binding probability by changing the average number of receptors and ligands that can form bonds (Note that if this number is considered fixed, as in Fig.2, nothing would change. It is only different when the binding probability is plotted against the grafting density). We take into account the effects of polydispersity by calculating the average binding probability given a certain number distribution for our sample, i.e.

$$\langle \theta \rangle = \int_0^{\infty} \theta(R)P(R), \quad (1)$$

where for the size probability distribution of our sample we assume a log-normal distribution of mean and variance as given by the TEM measurements of our samples:

$$P(R) = \frac{1}{\sqrt{2\pi}\sigma R} \exp \left[\frac{-(\ln R - \mu)^2}{2\sigma^2} \right]. \quad (2)$$

The mean $\bar{\mu} = \langle R \rangle$ and variance $\bar{\sigma}^2 = \langle R^2 \rangle - \langle R \rangle^2$ in our samples as measured by TEM were used to derive the parameters μ and σ for the lognormal distribution as:

$$\mu = \log \left(\frac{\bar{\mu}}{\sqrt{\frac{\bar{\sigma}^2}{\bar{\mu}^2} + 1}} \right) \quad (3)$$

and

$$\sigma^2 = \log \left(\frac{\bar{\sigma}^2}{\bar{\mu}^2} + 1 \right) \quad (4)$$

For the sake of providing a relevant example, we plot here the case of a monodisperse sample and a sample with a mean diameter of 23 nm and a mean-square root deviation of 12 nm, consistent with our samples, see Fig.1 here below. As somehow expected, the effect of polydispersity is to smooth the binding probability curve but the non-monotonic behaviour we dubbed range selectivity is again not much affected, despite the relatively high polydispersity in our polymersomes.

In the section “Colloidal stability of the polymersomes, polydispersity and its effect on range selectivity” of the Supplementary Information, we provide the same discussion, along with Fig.1 reported here (Fig.4 in the new SI) as well as all TEM data and their analysis.

Figure 1. Comparison of the binding probability of a mono vs polydisperse system. In the polydisperse system, we use a mean and variance equal to that of our sample at 1% ligand loading, which gives a value of the polydispersity index of approximately 0.25, representative to that of all our samples. As it can be seen, range selectivity is preserved and the effect of polydispersity is simply to slightly smooth the adsorption curve.

R: The author claims that they performed a confocal laser scanning microscope

for imaging. It is recommended to show some of the imaging data and corresponding analysis to emphasize the experimental findings and for in general, improve the article for readership.

In Fig.4 of the revised manuscript, Fig.2 reported here, we have now added three representative images taken via confocal laser scanning microscopy together with the binding data comparing theory vs experiments.

a)

b)

Figure 2. a) Adsorption probability as a function of ligands number, comparison of theory vs experimental data. Lines between datapoints are only a guide to the eye. The theoretical points have been calculated using the expression in Eq.(1), using a Poisson average over both the number of receptors and on the number of ligands, whose average value for the 1% loading of ligands (corresponding to a grafting density of ligands $\sigma_L/\sigma_{ref} = 1$), was used as a fitting parameter. **Size polydispersity of the particles was also taken into account, by using the experimentally measured mean and variance at each ligand grafting density, see the Supplementary Information.** Experimental error bars were calculated as an average over three independent measurements. See the Method section for more details on both the fitting procedure and the experimental measurements.

Figure 2. **b) Representative confocal microscopy images from which the experimental data on adsorption of polymersomes on cells were recovered. From left to right, polymersomes, cells and the composite image are shown. These images refer to one of the batches with relative grafting density of ligands equal to 1. All confocal microscopy data are available online in raw form from the DataFile.zip file**

We have further given more details about the specifics of the image analysis in a revised Methods section, which in its part related to the image analysis now reads:

Cells were left in Live Imaging Solution and the adsorption of polymersomes on cell membranes was analysed on a Leica SP8 confocal laser scanning microscope with 63X oil immersion lens. The total emission fluorescence of Cy₅-labelled polymersomes was measured in the 650 - 700 nm range using an excitation wavelength of 633 nm. Specifically, in every single test, for each formulation (corresponding to a given Angiopep2 percentage), the polymersomes fluorescence was collected by scanning the whole thickness of the cell using a z stack of 30 images. Every stack could include more than one cell and the fluorescence measurement per formulation was carried out on a minimum number of 40 cells. The total polymersome fluorescence of each stack was normalised by the number of cells and these normalised values were then averaged. All the experiments were done in triplicates. In order to make the image analysis automated, a custom-made script implemented on Matlab® was used (see the Supplementary Information).

The raw data from which the images reported in Fig.4 of the main text were generated are also provided in the DataFile.zip directly available on the publisher to allow access to others who would want to check our results. More in general, all data regarding all the images from which adsorption was measured have been stored on a local server and are made available upon request. This is because providing everything in a single data file is difficult given the size of the whole stack, which ranges in the TByte of data).

R: In Figure 2d, the energy vs. number of receptors plot, the number of receptors reaches up to 100. Even for repulsive interaction, polymer brushes (with or without ligand sites) need to make contact with the receptors on the surface.

A particle of the given size and grafting density, please calculate how many receptors will be in contact with the polymer chains — comment whether the surface is flat or bend, if so, how that may affect this number.

We are not 100% sure whether we have correctly understood the question of the referee but we will try to articulate our answer here. The theoretical curves plotted in Fig.2 report the binding probability (or the binding energy, in the case of Fig.2d) as a function of the receptors interacting with the particle. In this regard, for any potential particle size, there is always a certain value for the receptors grafting densities for which that number will be possible, albeit not necessarily experimentally achievable.

In our polymersomes, we do not know what the exact value of the ligands and receptors grafting densities are, which is in fact recovered from the fitting. We do this by

i) using directly in the fit the number of ligands/receptors interacting with the particle as a parameter (N_L^{fit} and N_R^{fit}) and

ii) using these numbers to estimate the grafting density, via $\rho_L = N_L^{\text{fit}}/A_{\text{int}}^{\text{np}}$ and $\rho_R = N_R^{\text{fit}}/A_{\text{int}}^{\text{surf}}$ (see Fig.5 in Supplementary Information for the geometry). As discussed in the previous answer, these formulae only provide an order of magnitude estimation. Nevertheless, this estimate is in line with previous experimental data on similar cells where the receptor grafting density was estimated by fitting using a different method.

As for the effect of curvature, in reporting the grafting density of receptors from the fitted number of interacting receptors we consider our surface flat and rigid. Generally speaking, for a curved surface we expect that if the curvature has opposite sign with respect to that of the particle (i.e., the particle sits in a pit), then more receptors will interact, whereas if the surface has the same sign (i.e., the particle sits on top of a hill), less receptors can interact. However, for a deformable surface each of these two cases can occur. This is because whether bending will lead to particle wrapping, or to the surface moving further away from the particle, will depend on whether or not the overall ligand-receptor-mediated interaction leads to an effective attraction or repulsion with respect to the particle's surface. It should be noted, however, that again this does not influence the mechanisms leading to range selectivity for the simple reason that curvature effects increase both the number of interacting receptors and the number of interacting ligands at the same time, because it is the contact area what increases, not the density of receptors or ligands.

On another note, we want to highlight the fact that the value at which the binding probability starts to decrease with increasing receptors depends on all the parameters in the system that affect both the binding and repulsive contributions. For example, this number will decrease as single bonds become weaker bonds or the repulsive polymer brush becomes denser, as shown in Fig.2 of the main text, so it does not need to be 100. Finally, we quote that for the polymersomes used in the experiments reported in Fig. 4 (where we vary the number of ligands at fixed number of receptors), taking their average measured size, the estimated number of interacting ligands is in the range [0-24] for the different grafting densities explored, which is fully compatible with this system considering previous independent measurements [1].

R: From the fitting of experimental data, it is found that polymer brushes are larger (8 nm) than the tether, which binds the ligand to the nanoparticle (6 nm). Therefore, I would expect an early repulsion between the polymer brush and the receptor before there is the formation of a ligand-receptor bond. Please comment on that.

It is exactly as the referee comments. If this was not the case, receptors would not need to penetrate the brush to bind the ligands and one would not be able to tune repulsion, and thus the location of the binding range, by playing with the brush density.

In general, having an early repulsion, before the ligand-receptor bond can form, means that there is a kinetic barrier to forming bonds. This will affect the kinetics of binding, but not the equilibrium behaviour which we describe here. It is thus possible that if the barrier is too strong, binding would not be observed before a certain experimental time-scale is reached. For our experiments, where binding is indeed observed in the experimental timescales, this was not a problem. In our revised manuscript, we have added a part before the conclusions where we describe the applicability of our results and potential caveats. This part now includes the following discussion on the effect of the brush along the lines described above:

Regarding the potential role of the brush, it should also be noted that although it allows to control repulsion, burying the ligand too deep inside it could lead to a significant kinetic barrier, affecting the timescale required to observe the equilibrium behaviour we describe. The kinetic barrier in this case stems from the fact that before a receptor can bind to

a ligand and recover part of the free-energy through bond formation, the brush must be compressed. In our experiments, a rough order of magnitude estimate gives a value for this timescale between 10^{-2} s and 10^{-1} s, well below the experimental timescale for the adsorption measurements of 1h. In general, however, the typical timescale depends on the nanoparticle concentration and on the total repulsive contribution F_{rep} (see the Supplementary Information).

R: On page 5, ‘ N_L should be interpreted as’ It must be written that N_R is also the number of receptors present in the contact region between the nanoparticle and surface.

We perfectly agree with the referee and we have now changed the main text to clarify this point:

N_L and N_R should be interpreted as the number of ligands and receptors, respectively, in the contact region between the nanoparticle and the binding site. In other words, these are the ligands and receptors that, given a certain nanoparticle orientation, can form bonds, and not their total number on the nanoparticle or adsorption site (see details in the Supplementary Information).

R: In equation (3), N_ϕ is the number of bonds between nanoparticle and surface receptors, which should be mentioned immediately after the equation. I assume that the authors consider the upper limit of $N_\phi = \min(N_L, N_R)$ instead of $\max(N_L, N_R)$.

The referee is perfectly right regarding N_ϕ and we now clarify its meaning immediately after Eq.3 is presented. We also thanks the referee for spotting the typo in the equation, as we clearly consider $N_\phi = \min(N_L, N_R)$ (as this minimum is actually the maximum number of bonds that can be made if one had N_L ligands and N_R receptors). This typo has now also been corrected.

R: In Figure 2a, b, c author should mention the value of the parameters which

are kept fixed, for example, in Figure 2a, the value of A used in the calculation.

The referee is right as without these values reproducibility of our results would not be ensured. We apologise for not putting these before and do it now in the revised form of the manuscript, adding the following sentence in the caption of Fig.2:

The various parameters which are fixed are chosen to show in each case a range where the non-monotonic behaviour is observed. The nanoparticle size, number of interacting ligands and activity, are kept fixed at $R = 50$ nm, $N_L = 3$, $z = 10^{-9}$ M, the other values used are $\sigma_0 = 1.95$ nm², $V_R = 110$ nm³, $\delta = 0.9$; $\beta\Delta G = -9$, $V_R = 40$ nm³, $L = 9$ nm and $\beta\Delta G = -9$, $\sigma_0 = 1.95$ nm², $\delta = 0.9$ in panel a), b) and c), respectively. Note that in panel b) we keep fixed the value of the length of the ligand rather than the insertion ratio δ . In all cases, we assume a zero contribution to the repulsive energy from the ligands, but its inclusion would only result in a rescaling of the activity z to lower values, from $z \rightarrow z \exp(-\beta F_{\text{rep}}^{\text{ligands}})$, which does not affect trends observed here.

R: In Figure 2d, changing the ΔG , at a fixed A , there is an insignificant change in the N_R at which the minima of F_{tot} appear. But in Figure 2a, why do we see the different saturation NR as a function of ΔG ? Could you repeat the plot of Figure 2d for the same parameter space used in Figure 2a?

The reason why one can observe different saturations as a function of N_R , despite the fact that the position of the minimum of F_{tot} appears to be at the same value of N_R (i.e., it is roughly independent of ΔG) is that observing saturation only requires F_{tot} to be below a certain value, but not necessarily at its minimum. This is because the binding probability is a sigmoidal curve and flattens rapidly to its saturation value of 1 as soon as $zq = zv_{\text{bind}} \exp(-\beta F_{\text{tot}}) > 1$. We also point out that the values of N_R required for saturation might look different from what can be inferred by looking at Fig.2d because of the effect of using different scales for the x-axis: logarithmic for Fig.2a,b,c, but linear for Fig.2d.

The different choice of scale (and the specific values of the parameters used in Fig.2d) was made for illustration purposes to clearly show the effect of changing ΔG and A . For this very same reason, we would prefer to keep the values used for Fig.2d as they are. However,

as the referee asked, we attach here Fig.3, a mock version of Fig.2d of the main manuscript using the parameters in Fig.2a of the main manuscript, as asked by the referee. In practice, Fig.3 plots the total interaction energy as a function of the number of receptors, as well as a related plot for the same combination of parameters but where we report q vs N_R (q being the partition function) rather than F_{tot} vs N_R . We use this second plot because we believe this better clarifies the reason behind the different ranges of saturation as a function of ΔG . We have also added the following sentence in the caption of Fig.2 to warn the reader about the different scales to avoid confusion.

Note that for illustration purposes the scales between Fig.2a)-2c) and Fig.2d) are different (logarithmic vs linear).

R: Mention the unit of A and ΔG in Figure 2d.

In the previous manuscript, we did not report any unit because A and ΔG were energies and, for consistency with what we had written in the main text (“energies are always considered to be scaled by the thermal energy $k_B T$, where k_B is Boltzmann’s constant and T temperature”), they were reported as if they were dimensionless quantities. We noticed however that some confusion may arise because we were not consistent throughout the paper as in the previous version of Fig. 2a we reported $k_B T$ as the units of ΔG . We have now changed the whole text and always report energies as $\beta \Delta G$ (or βF) to explicitly include the scaling by the thermal energy $k_B T = \beta^{-1}$. As currently written, “ A ” remains instead a dimensionless parameter.

R: I assume in Figure 2, the distance between nanoparticle and surface remains fixed. How do the authors choose the distance?

We have chosen this distance to be equal to the gyration radius of the ligand, because close to that region F_{tot} takes its lowest value and thus dictates the value of the bound partition function q . The reason for this choice is the following. A more general definition of the bound partition function q (see Eq.1 in the main text) can be done by calculating F_{tot} for different distances d consistent with the possibility to form bonds and including a distance-

Figure 3. a) Total adsorption free energy F_{tot} as a function of the number of receptors, plotted for the same curves as in Fig.2a) in the main manuscript. b) as in a), but in this case $q = \exp(-\beta F_{\text{tot}})$ is plotted rather than βF_{tot} . Note that the dashed horizontal line represents the value $q = z^{-1}$. c) A zoom of panel b) to the region where q crosses the line z^{-1} .

As it can be seen by comparison with Fig. 2a) in the main manuscript, saturation of the probability to its maximum value of 1 occurs as soon as the parameter $z \gg q^{-1}$, whereas it quickly drops to zero for $z \ll q^{-1}$.

dependent additional contribution $\Delta G_{\text{conf}}(d)$ to the single bond energy, see [2] and [3] in particular their Supporting Information, for details. In this way, q can then be calculated by integrating over all possible values of this distance. Given its exponential form, the value of this integral is well approximated via the maximum-term approximation / saddle-point approximation, using the distance corresponding to which F_{tot} is minimum.

This distance is approximately R_g (the gyration radius of the ligand) because of the following reasons: for $d \ll R_g$, the ligands become highly compressed against the binding surface, quickly increasing the repulsive contribution F_{rep} . For higher values, however, the repulsive

contribution becomes exponentially small. At the same time, for $d \gg R_g$ the ligands must stretch a lot to bind the receptors, leading to very weak bonds and in turns too small (in magnitude) values of F_{att} , the attractive part of the adsorption energy due to bond formation. For this reason, the minimum of $F_{\text{tot}} = F_{\text{rep}} + F_{\text{att}}$ must be around $d = R_g$. We have now added a section in the Supplementary Information (“**Calculation of the partition function q** ”) to explain this point in detail.

In equations 9 and 10, what is p ? I assume that p is the fraction of free ligand or receptor at a particular orientation. It means even at a certain number of N_R , there is a repulsive term due to free linkers or receptors? How do you incorporate it into the model?

The exact physical interpretation of p in Eq. 9 and 10, and more generally of the binding free-energy, is given in [2, 4] where those two equations were first derived. Here, we simply use them and cite those references in the main manuscript. Because this binding energy is given as the difference in the free-energy between a system where any allowed number of bonds can be made with respect to an equivalent system with the same position for the ligands/receptors but where no bonds are possible, the repulsive contribution can be calculated separately as the sum of the repulsive contributions from each ligand / receptor and simply added to F_{tot} , see again [2] for a detailed discussion of this point (also already included in the main manuscript).

The exact form of the repulsive energy for receptors and ligands only results in different values for the A and B coefficients describing the average repulsion per ligand and receptor, respectively (see Eq.4 and 6 of the main manuscript) and is model-dependent because it depends on the exact polymeric properties of ligands, receptors, the grafting surface, the brush and so on. In this regard, Eq.5 of the main manuscript, previously derived in [1], provides a formula for A for our specific experimental system. However, as we try to stress in the manuscript, the occurrence of range selectivity – our main and more general result - does not depend on these details.

R: Minor comments

R: Page 14, Last line “in Fig24...” should be “in Fig2 and Fig4 ..”

We thank the referee for spotting this typo, now corrected. It was supposed to be Fig.2-4

If you use V_R as the volume of the receptor, please use the same notation throughout the article. Same for other parameters.

Thanks again and apologies for this inconsistency. We have now double-checked the whole manuscript and corrected any previous inconsistency.

Reviewer 2 (Remarks to the Author): The authors propose a generic mechanism to generate what they call ‘range selectivity’ in the recognition of receptor densities on (cell) surfaces by multivalent nanoparticles (NPs). The mechanism is based on two opposing forces: attraction derives from ligands on the NP interacting with a few receptors on the surface, whereas repulsion derives from a repulsive brush (or any other repulsive medium) on the NP interacting with all receptors within the interaction range of the NP. The authors argue that the balance of attractive and repulsive interactions leads to a non-monotonous dependence of probe binding on receptor surface coverage with a maximum at intermediate coverage and gradually reduced binding towards lower and higher coverages. The idea certainly has its merit, and is potentially of interest to a broad audience. However, I have a number of concerns with this study which I am afraid will require extensive revision to address. Some of the points of critique listed below refer to the quality and comprehensiveness of the presented data, and others to the practical usefulness of the proposed approach as well as style.

We thank the reviewer for appreciating the merit of our idea. We answer to his/her questions point-by-point below and we recognize from the outset that a lot of the data and controls were simply omitted from the main manuscript for ease of reading but should have been given at least as Supplementary Information. We have now done this and we have also included all data file in the DataFile.zip file, as also requested from the editors.

R: The experimental data provided are very limited, making it impossible to assess the rigour of the experimental work and data analysis, and the relevance of the theoretical model for the experimental system. Amongst the questions that remain open are: (a) How did the authors confirm the ligand labelled polymers (i) mix well with the non-labelled and fluorescently labelled polymers and (ii) do not affect polymersome size? Both aspects are important to ascertain homogeneous dilution of the ligands within the polymersomes and that the polymer mixing ratio is indeed a reliable proxy of relative ligand density as the authors assume.

We are well aware of these typical problems (e.g., phase separation, clustering,...) arising in ligand-functionalised polymersomes, see e.g. [5] and [6] (now also cited in the Supplementary Information of the revised manuscript). This is not a problem in our system for the following reason. We work at a very low number of ligands, hence more than 90% of the polymersomes even at the highest ligand loading are made of pristine copolymers (i.e., non functionalised with a ligand). Because the ligand is attached to a copolymer whose only difference with the pristine one is the length of the hydrophilic block, we expect to be well within the miscibility region of the ligand-functionalised vs pristine copolymer. Moreover, even if this was not the case and the peptide induced phase separation or clustering, the demixing kinetics would be considerably slower (months) than our sample ageing, especially considering that our samples were stored at 4°C and were used immediately after their preparation. As for the fluorophore-labelled polymers, this is even more unlikely and even if it did, the concentration is the same across all the samples including the zero ligand formulation. Having said that, in the same references cited above it was shown that the thermodynamic tendency of block copolymers to phase separate and generate a not-homogeneous distribution of domains within the vesicle surface is evident when different polymers are used to form polymersomes and the result can be easily detected through confocal and electron microscopy techniques that reveal discontinuities in the topology of these polymeric nanovesicles. In this work, the polymers (identical in terms of chemical nature: Angiopep-2-PEGP-DPA, Cy5-PEG-PDPA and PEG-PDPA) and the protocol we have used to prepare them indeed allow to obtain well-dispersed ligands within the polymersomes without creating dissimilarities in the vesicle surface as confirmed by Transmission Electron Microscopy (TEM) analysis (see the new Supplementary Information). We have now added a section “Homogeneity of ligands surface distribution in our polymersomes” justifying our assumption as above in the Supplementary Information.

R: How monodisperse were the polymersomes, and have the authors considered how size-preferences in binding could bias the results?

We have indeed measured the PDI of our polymersomes by TEM and all the data are now reported in the Supplementary Information in the “Colloidal stability of the polymersomes, polydispersity and its effect on range-selectivity”. Whereas details can be found there, here

we quote that the average PDI in our sample, $PDI = (\mu/\sigma)^2$, μ being the average size and σ the square root of its variance, is around 0.25 in our samples, suggesting a relatively high polydispersity in our system.

Given the physics leading to its appearance, the non-monotonic binding behaviour that we dub range-selectivity is robust with respect to polydispersity in the sense that it will still be observed for polydisperse systems, mostly because it will occur independently of size. That this is the case is shown in Fig.4 here, where we compare a monodisperse sample and a polydisperse one with the same variance (and mean) to that of our experimental samples at 1% ligand loading, which give a PDI of ≈ 0.25 , common to all our samples. The size-probability distribution was assumed to be log-normal, which ensures only particles with positive radius are considered, as physically meaningful.

Figure 4. Comparison of the binding probability of a mono vs polydisperse system. In the polydisperse system, we use a mean and variance equal to that of our sample at 1% ligand loading, which gives a value of the polydispersity index of approximately 0.25, representative to that of all our samples. As it can be seen, range-selectivity is preserved and the effect of polydispersity is simply to slightly smoothen the adsorption curve.

The theoretical curve was calculated by noting that a particle's size affects, at a fixed surface density of receptors, the repulsive contribution as well as the number of interacting receptors. For each size, a different free-energy of adsorption F_{tot} and thus a different binding probability (Eq.1 in the main manuscript) occurs. For this reason, the effect of polydispersity can be introduced in the theoretical model by making a convolution of the binding probability function with the given size-distribution. It can be appreciated that polydispersity even at this relatively high level only slightly smoothens the adsorption curve but not much else. The same graph (and numerical details to obtain it) is now also reported and commented in the same section in the Supplementary Information.

We also point out that we now re-fitted all the experimental data taking into account the fact that different grafting densities of ligands lead to different sizes and different polydispersity level, resulting in a new Fig.4 and a new set of values for the fit parameters.

R: How did the nanoparticles distribute on the cell surfaces? Did the authors observe nanoparticle uptake in addition to binding, and if so, how did the authors discriminate between these two scenarios? Representative fluorescence micrographs of the polymersomes on cells need to be shown, and adequate analysis of the data demonstrated.

Representative fluorescence micrographs are now presented in the main text (along with a revised Fig.4) and more importantly we have also put all the material in raw form in the DataFile file as requested so that a full scrutiny can be easily made. In this regard, we would also like to underline that for the experimental procedure, the cells were fixed with paraformaldehyde (PFA) 3.7% (V/V) before incubation with polymersomes. The fixing process with PFA cross-links molecules by forming covalent chemical bonds between proteins and creating an insoluble mesh that preserves cellular architecture and composition, including the presence of receptors on the plasma membrane, and it also prevents any process of endocytosis. For this reason, we only took into account the binding of the polymersome on the cellular surface. We now added the following sentence in the Method section of the paper to discuss this point:

The fixing process with paraformaldehyde (PFA) cross-links molecules by forming covalent

chemical bonds between proteins and creating an insoluble mesh that preserves cellular architecture and composition, including the presence of receptors on the plasma membrane, and it also prevents any process of endocytosis. For this reason, in analysing our data we only took into account the binding of the polymersome on the cellular surface but not their internalisation.

R: I find the symmetry argument on page 11 (second paragraph) confusing. In order for the experimental model (in which the ligand density is varied) to be ‘symmetric’ to the theoretical model (in which the receptor density is varied), I would have expected that the repulsive brush should also switch location (from the nanoparticle in the theoretical model to the cell surface in the experiment). The free energies of insertion into the polymer brush would likely be quite different if the objects to insert and the polymers forming the brush were tethered to the same or different surfaces?

When we say the system is symmetric we do not mean that one can exchange N_R with N_L and obtain the same binding curve. What we mean is that the mechanism that generates range-selectivity applies equally when increasing the number of receptors at fixed number of ligands, or the number of ligands at fixed number of receptors, because the physics is the same: there is a different scaling between repulsive and attractive contributions, linear (or more generally power-law) vs logarithmic.

In the case of receptors binding the ligands in the polymersomes, compression of the protective brush is the main repulsive mechanism. This not only allows to generate high repulsive forces (high level of A in Eq. 4 in the main manuscript), but more generally the brush gives an experimentally tuneable parameter to change their magnitude and hence change the region where binding will occur. When considering instead ligands binding to a bare surface, i.e. without a brush, a repulsive contribution dependent on the number of ligands is still expected because of the confinement of the polymer tethering the ligand between the binding surface and the impenetrable core of the polymersome. Such a contribution, considering that the binding distance between the polymersome and the surface is of order R_g , the gyration radius of the ligand (please also refer to the answer to referee 1) will be small, estimated to be approximately $0.4k_B T$ (please see the “Fitting of experimental data” part in the Method

section of the revised manuscript) but still present and enough to induced a drop in the probability at higher ligand density, as shown in the experimental data. We also added a note when describing more in detail the repulsive contribution from the ligands

For generality, a repulsive term should also be included for ligands, for which we would thus have:

$$\beta F_{\text{rep}}^{\text{lig}} = BN_L. \quad (5)$$

The value of B depends on the exact repulsive mechanism at play and on the specifics of the system. For example, if the grafting surface where receptors reside is also covered by a polymer coating, B would have the same functional form as in Eq.5 (but with different parameters). We should note, however, that even in the absence of any brush a repulsion should always be expected from excluded volume effects arising from the need to confine the ligands (or receptors) between the surface of the nanoparticle and the grafting surface [2].

We have also re-written the related paragraph in the main text to more clearly explain what we mean by “symmetric behaviour”:

“Although we illustrated range selectivity describing the case of polymer-coated nanoparticles binding to a receptor coated surface, it should be clear that by symmetry such behaviour must also arise in the equivalent scenario where one studies the adsorption probability at fixed receptor numbers but varying the amount of ligands. This is because of the linear term in the repulsive energy as a function of the number of ligands, Eq.6. Although in this case the repulsion cannot be necessarily attributed to ligand insertion into a brush, we still expect a linear contribution due to the confinement of the ligands in the interacting region between the hard-core of the nanoparticle and the cell surface.”

Having said that we do not need to exactly determine the origin of the repulsive contribution (whose proper calculation would require knowledge of various parameters difficult to characterise, especially in a biological system) to expect range selectivity. One can simply rely on the fact that in any mean-field model, which is a good first-order approximation, a linear scaling of the repulsive force with the number of ligands occurs, and thus the effect

of repulsion can be lumped together in a scaling constant (A, B in Eq.4 and 6 of the main manuscript) that can be used for fitting. As range selectivity will occur as long as repulsion scales faster than logarithmically, this is enough to expect its occurrence regardless of the details of the underlying repulsive mechanism.

R: Related to the above it may also be worth providing the reader with information about the likely location of the ligand within the polymer brush as this should quite sensitively depend on the length of the tether to which the ligand is linked as compared to the length of the polymers forming the brush. Ideally, I would like to see a control experiment with ligands on a long tether and other polymers forming a short brush: in this scenario the reduction in binding at high ligand densities should disappear, or at least be much less pronounced!

As described in the main text, the ligand active site in our experimental system is at approximately 6nm from the polymersome hard surface, or in other words it is buried 2nm below the protective brush average height.

In our experimental platform, unfortunately, we can only control the properties of the polymersome (e.g., ligands length and brush characteristics), but not those of the cells, which are given. For this reason, we have measured adsorption as a function of the number of ligands, at fixed receptors concentration, rather than the opposite (as explained in the text, the problem is symmetric). Thus, in order to do the control experiment suggested we would need to control the repulsion between the ligands and the cell surface, and not between the cell receptors and the brush on the polymersomes. However, controlling the repulsion between ligands and the cell surface would require tuning the cell surface properties, which we cannot do. Note in this regard that tuning the ligand length would not work because the polymer-some - cell binding distance, which determines the confinement of the ligand and hence its repulsion, is a function of the length of the ligands itself and, to a first approximation, this length would simply increase in such a way that the repulsive contribution remains constant. Having said that, the experiment suggested by the referee could be ideally performed using a purely synthetic system. As the referee correctly suggests, its expected outcome should be to observe that a reduction in binding at high ligand densities should disappear (or, actually, be less pronounced and shift to much higher ligands densities, since there would still be repul-

sion without brush penetration due to confinement because of the polymersome hard-core). We believe that upon publication our findings will generate enough interest to see this and other experiments, or revision of previous results in light of our findings, performed. To this purpose, we have also added one more prediction that could be important for applications and could spur further interest (see below). However, we believe our main result to be the phenomenon discovered from the theoretical analysis, which should apply generally to various systems given its robustness (as also indicated by Ref. 3). The experiments were only a way to present a potential example rather than a systematic validation. In discussing our results, we now add the following sentences to discuss potential experiments / systems where our results could be applied and further tested to provide a fuller validation of the theory and its applicability:

For a further prediction:

Also connected to the development of applications, we note that often multivalent constructs have been developed not to target a specific receptor range but rather to simply increase the overall binding strength [7]. This in turn allows to decrease the detection limit of a specific multivalent target, for example, an analyte in solution, useful for diagnostic purposes [8]. Even in this case, our theoretical analysis suggests a general point that could be translated into a design principle: increasing the number of interacting ligands, or in other words the ligand grafting density, beyond a certain value can actually be detrimental to binding, even when the density is not large enough to foresee any potential negative cooperative interaction. Based on our mean-field model we can provide an upper bound to the optimal number of interacting ligands as (see details in the Supplementary Information):

$$N_L^{\text{optimal}} = \frac{N_R}{B} - \chi^{-1}, \quad (6)$$

where again B is the average repulsion per ligand as defined in Eq.6 and $\chi = \exp(-\beta\Delta G)$ which, multiplying by the (irrelevant, here) binding volume, is nothing but the binding constant of the given ligand-receptor pair. This formula predicts that for weak enough bonds (larger ΔG), the value of N_L^{optimal} can become negative. More precisely, this means

that, in this case, the binding strength is a monotonically decreasing function of the number of ligands. This is a possibility that is seldom, if ever, considered when discussing the design of multivalent constructs using weak ligand receptor pairs. It is in fact somehow generally assumed that a higher number of ligands will always yield a higher binding strength. As we show here, this might not be the case as it does not account for the fact that more ligands also bring more repulsion, which might or might not be counterbalanced by the increase in the attractive contribution to the binding free energy. The same expression also shows that in the other limit, for very strong ligands ($\chi \rightarrow \infty$), the optimal number of ligands is still finite (with the caveat that $N_L > N_R$) and depends on the number of receptors as well as the strength of the repulsion, i.e. $N_L^{\text{optimal}} = \frac{N_R}{B}$.

For systems that can be used to a more systematic study / validation of our findings:

This latter prediction, as well as some of the trends presented in Fig.2, are difficult to test in a systematic way within our experimental system. However, we hope these results will spur further interest towards this goal. In this regard, we suggest that using fully synthetic systems where binding is mediated by ligand-receptor interactions, e.g., DNA-coated colloids and surfaces, would provide the perfect platform to further test these results in a more controlled manner.

R: Some critical analysis of the results of the fit would be worthwhile. Did the authors attempt to directly measure (or estimate) the receptor surface density on the cells to check if the derived number for ρ_R is reasonable. Likewise, is the size and organisation of the polymersome consistent with a scenario of 7 interacting ligands within the polymersome area?

We agree with the referee on this and have now added the following sentence to the part where we described the results of the fitting procedure, which now also considers the effects of polydispersity.

The overall procedure produced fitting parameters of $\sigma_L = 2 \cdot 10^{-2}/\text{nm}^2$ or between [0 - 24] ligands at the grafting densities considered. The grafting density was estimated by

fitting the number of interacting ligands and then assuming that all ligands grafted within a distance of $2R_G$ from the binding site were available for binding and that the equilibrium distance between the surface of the nanoparticle and that of the binding site was R_G , which thus provide an estimate of the interacting area on the polymersome of $A_{\text{int}}^{\text{pp}} = 2\pi R_{\text{np}} R_G$ [9] (see Fig.5 in the Supplementary Information for clarity). The fitted density of receptors was $\rho_R = 1.5 \cdot 10^{-4}/\text{nm}^2$, or approximately 10^{-2} receptors per interacting area on the adsorption site depending on the polymersome size, estimated as the projection of $A_{\text{int}}^{\text{pp}}$ on the flat adsorption surface and thus equal to $A_{\text{int}}^{\text{surf}} = \pi [R_{\text{np}}^2 - (R_{\text{np}} - R_G)^2]$. For what concerns the grafting density of ligands on the polymersomes and receptors on the cell membrane, the fitted values we obtain are within an order of magnitude, and thus consistent with, given our coarse-grained description, previous independent estimates obtained for the same system of $\approx 1.3 \cdot 10^{-3}/\text{nm}^2$ and $2.1 \cdot 10^{-5}/\text{nm}^2$, see [1].

Furthermore, what was previously a fitting parameter, the repulsion per ligand, has now been estimated using the model discussed in [10], which provided a value of $0.4k_{\text{B}}T$. Notably, keeping this value as a free parameter would have given a best fit of $0.5k_{\text{B}}T$. Here is the additional discussion we added in the Method section when discussing how we fit the data:

Considering that the ligand repulsion should be due to its interaction with the impenetrable surface of the cell, an estimate for B can be obtained by assuming the ligand behaves as a Gaussian chain at a distance R_G from a flat plane, using the formulas and parameters reported in [10], which gives: $B = a \exp(-b(r/R_g - c))$, with $a = 3.1995, b = 4.1662, c = 0.4996$ (these parameters were fit in [10] to reproduce exact Monte Carlo data for repulsion up to $10k_{\text{B}}T$), which for $r = R_G$ gives 0.40.

Finally, we also want to take this chance to point out that revising our manuscript we have found a small error in one of the formulas used in our code to estimate the interacting area - there was a missing factor of 2π , irrelevant for the consequences of the paper but which changes the quantitative values reported. Moreover, we are now including in the fitting the effects of polydispersity and we have estimated the repulsion due to ligands using the model in [10] rather than treating it as a fitting parameter (keeping it free would yield an optimal value of $0.5k_{\text{B}}T$ vs the estimate of $0.4k_{\text{B}}T$ we use here and the previous estimate, without accounting for polydispersity and the change in polymersome size at different grafting den-

sities, of $0.6k_B T$). For this reason, we have repeated the whole fitting procedure, leading to slightly different values than those previously obtained. The final values for ρ_R and ρ_L remain well within an order of magnitude of the previously fitted ones, a difference we would expect given that the estimates available via such coarse-grained models for the interactions are only supposed to give a rough estimate of these parameters.

R: Did the authors test if receptor mobility affects nanoparticle binding? This question would appear important, since the authors state in their manuscript (page 11) that lateral mobility should be suppressed for their model to work.

In our experimental system, receptor mobility is suppressed by the fixing procedure that cross-links all proteins. Having said that, we take this chance to clarify the point about mobility. Mobility does not need to be completely suppressed to observe range-selectivity. Rather, if mobility is present we need to consider an additional contribution to the free-energy, see Mognetti et al in [11], which complicates the quantitative analysis. However, on a qualitative level, mobility should only shift the drop in binding to higher values of grafting densities due to an additional contribution to the free-energy of the system required to confine a certain number of receptors within the polymersome-binding site interacting area $A_{\text{int}}^{\text{surf}}$ (see Fig.5 in the new supplementary info for clarity). Before the conclusions, we have now added a few paragraphs to the revised manuscript where we discuss details, limitations and applicability of our findings (please see the whole part before the conclusions), and we have now added the following paragraph to discuss specifically the effect of mobility:

Having said this, not all systems where binding depends on ligand-receptor bond formation obey the exact same physics described here and an extension of our model might be required. In this regard, an important point worth discussing is that of the mobility of ligands and receptors. Here, we limited our modelling to systems where both ligands and receptors are fixed. With some prominent exception [12, 13], this is true for most synthetic multivalent constructs, including functionalised colloids, nanoparticles or polymers. It is also true for certain biological targets. For example, our model would apply to the case where one aims to use ligand-functionalised nanoparticles to target membrane cellular receptors that are cross-linked to the underlying cytoskeleton, and viruses either with fixed receptors or where

the receptor density is so high that, although collective moves are possible, the local density of receptor is likely to be approximately fixed [14], e.g., in the coating of the influenza virus. In all these cases, our theoretical framework can be directly applied. When receptor (or ligand) mobility is high and can induce local changes of their grafting density, instead, a theoretical analysis of the binding mechanism should include its effects. Whereas a full description goes beyond the scope of the present manuscript, we notice that this could be done using some recent results derived by Moggetti *et al* in [11]. In practice, we expect range-selectivity to still occur but the drop of the binding probability at high densities of receptors (or ligands) to shift to higher values compared to those for the fixed case.

R: Whilst not all of the above-requested data may fit into the main manuscript it can readily be provided as supplementary information. Furthermore, it would be desirable to see theoretical data displayed in a more consistent and complete manner: (a) The shape of the curves in Figures 2 and 4, for example, looks quite different. Is this entirely due to the different x axes scaling (i.e. log vs linear) or are there further differences that are difficult to appreciate with the heterogeneous axis scaling?

We agree that the different axes scaling (logarithmic in one case, linear in the other) might make the different curves look more different than they actually are, for example, the long skew at low receptor / ligand number is purely an effect of it . However, in presenting the experimental data, we chose a linear scaling simply because we can only vary the number of ligands by a factor of 10 (and not a 1000 – or more – like in the presentation of theoretical analysis), which we believe is visualized better in this way.

Please note in this regard that in the previous theoretical fitting there was also another difference between Fig.2 and 4, which we explicitly hinted at but decided not to discuss: the presence of “steps” in adsorption probability in Fig.4 rather than a smooth curve as in Fig.2. The presence of these steps, or lack thereof, is system dependent and only occur for certain combination of parameters. Thus, these features do not constitute as general and robust a property as range selectivity, whose presentation to the readers is our goal in this paper. In fact, because of the inclusion of polydispersity effects, this step-like appearance can disappear (see for example the comparison between mono and polydisperse system in Fig.4). As we

believe this discussion does not add anything to the main findings, and we now only estimate adsorption explicitly at the experimental points rather than trying to interpolate between them in Fig.4 (because to do that one would need to know how the grafting density affects polydispersity, as we see in our system), unlike in the previous version of the manuscript, we have removed any reference to it.

R: The authors highlight superselectivity in their introduction; a discussion would be worthwhile on how the range selectivity proposed here relates to superselectivity and/or if the two can co-exist. For example, superselectivity is conveniently defined by the parameter $\alpha = d \log \theta / d \log \langle N_R \rangle$ exceeding 1; can a similar analysis be performed for range selectivity? To appreciate the co-existence of range selectivity and superselectivity it would be practical to see the data in Fig. 2a-c in log-log plots.

Although both super-selectivity and range selectivity can both occur in multivalent systems, the two are completely independent phenomena. In other words, if the conditions for super-selectivity to be observed are present (weak and multiple bonds, with a low bulk density / activity of particles), this will be observed, regardless of range selectivity. To keep the focus of the paper on range selectivity, we would prefer not to include any other extensive analysis on their relation here. It should also be pointed out that somehow range selectivity is more general than super-selectivity in the sense that it can occur for conditions (strong bonds, high bulk density of particles and even monovalent binding) where super-selectivity cannot be observed. In the additional discussion of our results now present in the revised manuscript before the conclusion, we have added the following paragraph discussing the relation between the two phenomena:

*Under specific conditions, multivalent constructs can display so-called super-selective binding [1, 15, 16], i.e. a sharp (superlinear) response to receptor density gradients. Although as we show here multivalent constructs can *also* display range selectivity, the two properties are independent. In fact, range selectivity is somehow a more robust, general phenomenon in the sense that it requires less stringent conditions to be observed. For example, it occurs also for monovalent constructs and does not require weak bonds, as necessary to observe super-*

selectivity. Thus in general, if the conditions for super-selectivity are met, this phenomenon can be observed together with range selectivity, whereas the opposite is not necessarily true.

R: Could Fig. 2b include an example of very high σ_0 to illustrate the critical importance of the repulsive brush for the reduced binding in the limit of at high $\langle N_R \rangle$

In the revised Fig.2b), we have now included the case of an “infinite” value of σ_0 (i.e. zero grafting density for the brush) to illustrate this point.

(d) Should the label on the x axes in Fig. 2 be $\langle N_R \rangle$ throughout?

Not exactly. It is indeed $\langle N_R \rangle$ for Fig.2a),b) and c) because the adsorption probability there is calculated via Eq.1 using a Poisson average for the number of interacting receptors centred around the given value of $\langle N_R \rangle$. However, in Fig.2d the free-energy is indeed calculated for a system where the number of receptors per binding site is kept constant, hence the difference in notation. Going from the free energy (i.e., Fig. 2d)) to the probability (i.e. Fig. 2a)-c) means calculating the free-energy for different fixed values of N_R , calculating the corresponding adsorption probability and then weighting this probability for the (Poisson) probability that such number of receptors is observed, a procedure that generally smooths the adsorption curve and practically increases the range of adsorption.

R: What were NL, Rnp and L for the example curves shown in Fig. 2?

These details are now provided for reproducibility in the revised caption of Fig. 2. We also mention that, as requested by the editors, we include the Python code used to generate these data and makes it available to readers.

‘Range selectivity’ appears entirely justified as a term for the reported phenomenon. I notice, however, that the effective binding ranges are generally quite large, typically spanning more than one order of magnitude in receptor density (an exception is the solid blue line in Fig. 2b). The authors may want

to comment on this as it would appear quite important a consideration from the perspective of practical utility.

We have now added the following paragraph to discuss this point, where we also add a further point regarding the consequences of range selectivity for applications.

In order to implement our results for the development of applications, e.g., for targeted drug delivery, some fine-tuning of the system is also required. For example, for selective drug-targeting one might want to restrict binding to a relatively small range of receptor densities. In the various examples presented in Fig.2, this range varies between about 2 and 3 orders of magnitude, which might be too large. However, we point out that we have made no efforts trying to optimise it. For example, the results in Fig.2 suggest that the higher the repulsion per receptor (ligand), the smaller the range where binding occurs. More generally, optimisation should be done within the experimental constraints by using all the available parameter space, which could be done with known minimization algorithms see, e.g., [17]. This includes the number of ligands or their binding constant, as well as characteristics of the repulsive brush such as the grafting density or degree of polymerisation (σ_0^{-2} and N in Eq.5).

Also connected to the development of applications, we note that often multivalent constructs have been developed not to target a specific receptor range but rather to simply increase the overall binding strength [7]. This in turn allows to decrease the detection limit of a specific multivalent target, for example, an analyte in solution, useful for diagnostic purposes [8]. Even in this case, our theoretical analysis suggest a general point that could be translated into a design principle: increasing the number of interacting ligands, or in other words the ligand grafting density, beyond a certain value can actually be detrimental to binding, even when the density is not large enough to foresee any potential negative cooperative interaction. Based on our mean-field model we can provide an upper bound to the optimal number of interacting ligands as (see details in the Supplementary Information):

$$N_L^{\text{optimal}} = \frac{N_R}{B} - \chi^{-1}, \quad (7)$$

where again B is the average repulsion per ligand as defined in Eq.6 and $\chi = \exp(-\beta\Delta G)$ which, multiplying by the (irrelevant, here) binding volume, is nothing but the binding constant of the given ligand-receptor pair. This formula predicts that for weak enough bonds (larger ΔG), the value of N_L^{optimal} can become negative. More precisely, this means that, in this case, the binding strength is a monotonically decreasing function of the number of ligands. This is a possibility that is seldom, if ever, considered when discussing the design of multivalent constructs using weak ligand receptor pairs. It is in fact somehow generally assumed that a higher number of ligands will always yield a higher binding strength. As we show here, this might not be the case as it does not account for the fact that more ligands also bring more repulsion, which might or might not be counterbalanced by the increase in the attractive contribution to the binding free energy. The same expression also shows that in the other limit, for very strong ligands ($\chi \rightarrow \infty$), the optimal number of ligands is still finite (with the caveat that $N_L > N_R$) and depends on the number of receptors as well as the strength of the repulsion, i.e. $N_L^{\text{optimal}} = \frac{N_R}{B}$.

The title does not appear consistent with the work shown. Firstly, the phenomenon described does not appear to require multivalent constructs, as shown for the ‘indifferent binding’ scenario which effectively refers to monovalent constructs. Secondly, whilst the repulsive interaction may be of some entropic nature (as for a polymer brush) one could equally well envisage some non-entropic force to play the repulsive part in the ‘game’.

The referee is correct in saying that range-selectivity does not require multivalent constructs and the example of indifferent binding is spot on. What we wanted to highlight with the title, however, is that this phenomenon will generally be observed for particles / constructs with any number of ligands or receptors, thus including also monovalent binders, because of the way the number of binding configurations, i.e. entropy, grows in the different receptor density regimes, $N_L \gg N_R$ vs $N_L \ll N_R$.

Although for this reason we believe the title to be acceptable, we are certainly not married to it and we are ready to change it if still deemed necessary. In this regard, we suggest the following new title: “A game of entropy: range selective binding in ligand receptor interactions”

The authors may want to check the literature for examples of experimental systems in which non-monotonous binding of soluble analytes as a function of surface-binding sites is observed. Steric hindrance at high densities of surface binding sites leading to reduced analyte binding has previously been recognised in solid-phase binding assays even for simple analytes such as (monovalent as well as multivalent) proteins. Although in this context the phenomenon is typically considered a nuisance rather than a benefit, it bears the signature of range selectivity.

We were aware of that literature but the microscopic mechanism for the non-monotonic binding used to justify these previous observations, steric hindrance between receptors on the surface which leads to a decrease in the binding constant for a single ligand-receptor bond (anti-cooperativity) is completely different from the one presented here. In fact, “our” version of range selectivity would arise also in the case where the single ligand-receptor interaction is completely independent from the receptors densities, as we consider here. In any case, we agree that that literature present another type of range-selectivity and we now cite here a comprehensive review on solid-state binding assays discussing this phenomenon. In the revised main text, we also add the following paragraph to discuss the difference between the two mechanisms:

On a different note, we point out that we are not the first to describe a decrease in binding at high receptors concentration, a phenomenon that has long been recognised and discussed in the literature, in particular in solid-phase binding assays for proteins, see, e.g., [18] and references therein. However, the mechanism attributed to non-monotonic binding in these experiments is completely different from what we describe here. In those cases, it is assumed that at high receptors density the proximity between different receptors leads to limiting the accessibility of their binding sites for the ligands. This is equivalent to making the single-bond strength depending on density and decreasing in an anti-cooperative way for increasing densities. For this reason, this phenomenon can only occur for very high grafting densities of receptors and regardless of any property of the bound construct. In our case, no cooperative effect is invoked (ΔG is assumed independent of receptor/ligand density) and as we show it is the relative number of ligands vs receptors what matters because this dictates

the growth of the entropy of binding in the system. When we account for this, we still conclude that a decrease in adsorption probability must occur due to the different scaling of bond-mediated attraction and repulsion. Importantly, we also show that the mechanism we describe here can be tuned by changing the properties of the binding construct, independently of the targeted surface.

Reviewer 3 (Remarks to the Author): Meng Liu and co-authors present a manuscript demonstrating range-selective binding in multivalent binding scenarios. They derive theoretically why, for a very broad class of systems (some requirements are specified but they are very mild and most systems would satisfy them) it should be expected that binding is most prevalent at an intermediate range of receptor densities, and not below (obviously) or above (much less obvious). This would have been publishable by itself. Furthermore, the authors add data from an experimental system involving polymersomes and cancer cells that demonstrates the predicted behavior. These are very important findings that will interest a broad readership from medical biologists to physicists. I therefore recommend this manuscript for publication. The main result is theoretical and does not involve any complex calculation methods, so reproducibility is not an issue (I cannot assess how hard it would be to reproduce the experimental part).

We thank the reviewer for the very positive and supportive comments. We perfectly agree with him/her that the beauty of these results depends on their generality and very minor assumptions required.

R: Of course, I do have some recommendations that I would like the authors to consider before making the recommendation final. It is not clear how equations 4 and 5 are derived. I understand technical details are not desirable in the main text of a letter of this type, but perhaps a few words could be added, or a methods section similar to section A.

We agree with the reviewer and now direct the reader towards the new Supplementary Information, where all the details are given in the section “A mean-field approximation to the radial binding scenario”. Whereas all the details can now be found there, in practice, we just point out that those results are derived by writing the bound partition function (i.e. the sum over states where at least one bond is present) for either N_L non-competing ligands, each of which can bind N_R receptors (Eq. 4), or for N_R independent receptors that can bind N_L ligands in the opposite case (Eq.5), hence the symmetry. The “independent ligand(receptor)” approximation works because when the number of ligands is much smaller than the number

of receptors, the number of configurations for which one bound ligand prevents a second one to be also bound to the same receptor is a small fraction of the total number of bound configurations (the same for symmetry is true in the case where it is the number of ligands that is much higher than that of receptors). Hence, it is close to a system where ligands were truly independent from each other. A more formal / less hand waving argument is given in [4] and we also redirect the reader to it for a more in-depth discussion.

R: Methods section A refers to figure 24, which does not exist (should be 3).

We thanks the reviewer for spotting this typo, it should have been Fig. 2-4. We have now corrected it in the main manuscript.

R: A restriction on the range of applicability of the results is detailed on page 11 (about the receptors having to be either fixed in space or laterally immobile). I feel this comes a bit early (it feels as if it should be part of the discussion, but the paragraph after it still contains results), and therefore does not get the attention it needs. Perhaps the authors can add a statement on how generally this requirement is satisfied, and name a few systems for which it might not be? I would also reconsider the order of presentation but I leave this optional because the letter reads quite nicely as it is.

We thank the reviewer for this comment and we now take this chance to revise the manuscript to discuss and expand on the applicability of the results we present, their limitations and possible examples of systems where range selectivity can thus be observed. The cited paragraph has now been moved within this longer discussion, which we report here in full:

In order to better understand the range of applicability of our results, and their possible implementation in other systems, we would like to discuss a few details and limitations of our approach, highlighting in particular those areas where application of the concepts presented here might need extra care. In this regard, we start by pointing out that although we discuss here the problem from the perspective of a multivalent particle binding to a surface, a similar physical picture would more generally apply for the ligand-receptor-mediated binding

of two multivalent constructs. This includes binding of a nanoparticle or polymer to a virus, for example, as in the development of antiviral applications [19, 20], or binding of a nanoparticle to a cell, like in the experiments we present here. Having said this, not all systems where binding depends on ligand-receptor bond formation obey the exact same physics described here and an extension of our model might be required. In this regard, an important point worth discussing is that of the mobility of ligands and receptors. Here, we limited our modelling to systems where both ligands and receptors are fixed. With some prominent exception [12, 13], this is true for most synthetic multivalent constructs, including functionalised colloids, nanoparticles or polymers. It is also true for certain biological targets. For example, our model would apply to the case where one aims to use ligand-functionalised nanoparticles to target membrane cellular receptors that are cross-linked to the underlying cytoskeleton, and viruses either with fixed receptors or where the receptor density is so high that, although collective moves are possible, the local density of receptor is likely to be approximately fixed [14], e.g., in the coating of the influenza virus. In all these cases, our theoretical framework can be directly applied. When receptor (or ligand) mobility is high and can induce local changes of their grafting density, instead, a theoretical analysis of the binding mechanism should include its effects. Whereas a full description goes beyond the scope of the present manuscript, we notice that this could be done using some recent results derived by Mognetti *et al* in [11]. In practice, we expect range-selectivity to still occur but the drop of the binding probability at high densities of receptors (or ligands) to shift to higher values compared to those for the fixed case.

Another relevant question is related to the repulsion required to observe range selectivity. In principle, any force that grows with the number of receptors (or ligands) faster than logarithmically would be enough. To the best of our knowledge, this includes all known repulsive mechanisms. Practically, however, one needs repulsion to overcome attraction at an experimentally achievable receptor (ligand) density to observe the drop in binding. This can be understood based on the results in Fig.2 d), which show that the smaller the repulsion per receptor (ligand), the larger the range where binding will occur, which could shift the number of required receptors to observe a drop in binding to physically inaccessible values. In this regard, although here we suggest to use a polymer brush because it provides a highly-tuneable parameter to control repulsion and observe the non-monotonic behaviour within a specific region, its presence is not a strict requirement. Even in its complete absence,

the small ($\approx k_B T$) repulsion due to ligands or receptors being confined within the binding region between the surface of the nanoparticle and that of the adsorption site (see Fig.5 of the Supplementary Information), which limits their allowed microscopic configurations causing steric repulsion [2], might be enough. For example, this is the mechanism we invoke to justify the decrease in binding with the number of ligands in our experiments.

Regarding the potential role of the brush, it should also be noted that although it allows to control repulsion, burying the ligand too deep inside it could lead to a significant kinetic barrier, affecting the timescale required to observe the equilibrium behaviour we describe. The kinetic barrier in this case stems from the fact that before a receptor can bind to a ligand and recover part of the free-energy through bond formation, the brush must be compressed. In our experiments, a rough order of magnitude estimate gives a value for this timescale between 10^{-2} s and 10^{-1} , well below the experimental timescale for the adsorption measurements of 1h. In general, however, the typical timescale depends on the nanoparticle concentration and on the *total* repulsive contribution (see the Supplementary Information). In order to implement our results for the development of applications, e.g., for targeted drug delivery, some fine-tuning of the system is also required. For example, for selective drug-targeting one might want to restrict binding to a relatively small range of receptor densities. In the various example presented in Fig.2, this range varies between about 2 and 3 orders of magnitude, which might be too large. However, we point out that we have made no efforts trying to optimise it. For example, the results in Fig.2 suggest that the higher the repulsion per receptor (ligand), the smaller the range where binding occurs. More generally, optimisation should be done within the experimental constraints by using all the available parameter space, which could be done with known minimization algorithms see, e.g., [17]. This include the number of ligands or their binding constant, as well as characteristics of the repulsive brush such as the grafting density or degree of polymerisation (σ_0^{-2} and N in Eq.5).

Also connected to the development of applications, we note that often multivalent constructs have been developed not to target a specific receptor range but rather to simply increase the overall binding strength [7]. This in turn allows to decrease the detection limit of a specific multivalent target, for example, an analyte in solution, useful for diagnostic purposes [8]. Even in this case, our theoretical analysis suggest a general point that could be translated into a design principle: increasing the number of interacting ligands, or in other words

the ligand grafting density, beyond a certain value can actually be detrimental to binding, even when the density is not large enough to foresee any potential negative cooperative interaction. Based on our mean-field model we can provide an upper bound to the optimal number of interacting ligands as (see details in the Supplementary Information):

$$N_L^{\text{optimal}} = \frac{N_R}{B} - \chi^{-1}, \quad (8)$$

where again B is the average repulsion per ligand as defined in Eq.6 and $\chi = \exp(-\beta\Delta G)$ which, multiplying by the (irrelevant, here) binding volume, is nothing but the binding constant of the given ligand-receptor pair. This formula predicts that for weak enough bonds (larger ΔG), the value of N_L^{optimal} can become negative. More precisely, this means that, in this case, the binding strength is a monotonically decreasing function of the number of ligands. This is a possibility that is seldom, if ever, considered when discussing the design of multivalent constructs using weak ligand receptor pairs. It is in fact somehow generally assumed that a higher number of ligands will always yield a higher binding strength. As we show here, this might not be the case as it does not account for the fact that more ligands also bring more repulsion, which might or might not be counterbalanced by the increase in the attractive contribution to the binding free energy. The same expression also shows that in the other limit, for very strong ligands ($\chi \rightarrow \infty$), the optimal number of ligands is still finite (with the caveat that $N_L > N_R$) and depends on the number of receptors as well as the strength of the repulsion, i.e. $N_L^{\text{optimal}} = \frac{N_R}{B}$. This latter prediction, as well as some of the trends presented in Fig.2, are difficult to test in a systematic way within our experimental system. However, we hope these results will spur further interest towards this goal. In this regard, we suggest that using fully synthetic systems where binding is mediated by ligand-receptor interactions, e.g., DNA-coated colloids and surfaces, would provide the perfect platform to further test these results in a more controlled manner.

We would like now to ultimately discuss our results in relation to other peculiarities of multivalent binding, as well as to draw a distinction between range selectivity as described here and non-monotonic binding reported in literature for protein or antibody/antigen binding [18]. Multivalent constructs can display, under specific conditions, so-called super-selective binding [1, 15, 16], i.e. a sharp (superlinear) response to receptor density gradients. Here we

show that multivalent constructs can *also* display range selectivity but the two properties are independent from each other. In fact, range selectivity is somehow a more robust, general phenomenon in the sense that it requires less stringent conditions to be observed. For example, it occurs also for monovalent constructs and does not require weak bonds, as necessary to observe super-selectivity. Thus in general, if the conditions for super-selectivity are met, this phenomenon can be observed together with range selectivity, whereas the opposite is not necessarily true. On a different note, we want to point out that we are not the first in describing a decrease in binding at high receptors concentration. This phenomenon, in fact, has long been recognised and discussed in the literature, in particular in solid-phase binding assays for proteins, see, e.g., [18] and references therein. However, the mechanism attributed to non-monotonic binding in these experiments is different from what we describe here. In those cases, it is assumed that at high receptors density the proximity between different receptors leads to limiting the accessibility of their binding sites for the ligands. This is equivalent to making the single-bond strength depend on density, more precisely, decreasing in an anti-cooperative way for increasing densities. For this reason, this phenomenon can only occur for very high grafting densities of receptors and is independent from the properties of the bound construct. In our case, no cooperative effect is invoked (ΔG is assumed independent of receptor/ligand density) and it is the relative number of ligands vs receptors what matters because it dictates the way the entropy of binding grows in the system. When we account for this, we still conclude that a decrease in adsorption probability must occur due to the different scaling of bond-mediated attraction and repulsion. Importantly, we also show that the mechanism we describe here can be tuned by changing the properties of the binding construct, independently of the targeted surface.

R: Equation 7 confuses me: I would think one should either sum over all values of N_ϕ and include the Ω , or sum over all configurations ϕ and leave out the Ω . The main text chooses the former. This equation seems to have them mixed up. Please clarify the notation and (if you agree that it's currently not consistent) correct the equation.

The referee is perfectly right and we thank him/her for spotting this inconsistency. Because what is crucial here is Omega / Entropy, we now show that the sum is over all possible

values of N_ϕ and include $\Omega(N_\phi)$ in the equation.

R: The legends in figure 2 are very small and hard to read.

We have now made all labels larger and improved the graphs readability

Finally, we want to take the chance provided by this revision to correct another typo in the reported formula for the probability associated to the binding free-energy (see Eq 12 in the main manuscript), which should read:

$$p_L = \frac{(N_L - N_R) \chi - 1 + \sqrt{4N_L\chi + (1 + (N_R - N_L)\chi)^2}}{2N_L\chi}$$

- [1] Xiaohe Tian, Stefano Angioletti-Uberti, and Giuseppe Battaglia. On the design of precision nanomedicines. Science Advances, 2019.
- [2] P. Varilly, S. Angioletti-Uberti, B. M. Mognetti, and D. Frenkel. A general theory of dna-mediated and other valence-limited colloidal interactions. The Journal of Chemical Physics, 137:094108–094122, 2012. doi:http://arxiv.org/abs/1205.6921.
- [3] Tine Curk, Jure Dobnikar, and Daan Frenkel. Optimal multivalent targeting of membranes with many distinct receptors. Proceedings of the National Academy of Sciences, 114(28): 7210–7215, 2017.
- [4] S. Angioletti-Uberti, P. Varilly, B. M. Mognetti, A.V. Tkachenko, and D. Frenkel. Communication: A simple analytical formula for the free energy of ligand-receptor-mediated interactions. The Journal of Chemical Physics, 138:021102–021106, 2013.
- [5] Caterina LoPresti, Marzia Massignani, Christine Fernyhough, Adam Blanzaz, Anthony J Ryan, Jeppe Madsen, Nicholas J Warren, Steven P Armes, Andrew L Lewis, Somyot Chirasatitsin, et al. Controlling polymersome surface topology at the nanoscale by membrane confined polymer/polymer phase separation. ACS nano, 5(3):1775–1784, 2011.
- [6] Lorena Ruiz-Pérez, Lea Messenger, Jens Gaitzsch, Adrian Joseph, Ludovico Sutto,

- Francesco Luigi Gervasio, and Giuseppe Battaglia. Molecular engineering of polymer-some surface topology. Science Advances, 2(4), 2016. doi:10.1126/sciadv.1500948. URL <https://advances.sciencemag.org/content/2/4/e1500948>.
- [7] Mathai Mammen, Seok-Ki Choi, and George M Whitesides. Polyvalent interactions in biological systems: implications for design and use of multivalent ligands and inhibitors. Angewandte Chemie International Edition, 37(20):2754–2794, 1998.
- [8] Carlo Fasting, Christoph A Schalley, Marcus Weber, Oliver Seitz, Stefan Hecht, Beate Kokschi, Jens Dornedde, Christina Graf, Ernst-Walter Knapp, and Rainer Haag. Multivalency as a chemical organization and action principle. Angewandte Chemie International Edition, 51(42):10472–10498, 2012.
- [9] Vladimir P Zhdanov and Fredrik Höök. Diffusion-limited attachment of large spherical particles to flexible membrane-immobilized receptors. European Biophysics Journal, 44(4):219–226, 2015.
- [10] Francisco J Martinez-Veracoechea, Behnaz Bozorgui, and Daan Frenkel. Anomalous phase behavior of liquid–vapor phase transition in binary mixtures of dna-coated particles. Soft Matter, 6(24):6136–6145, 2010.
- [11] Bortolo Matteo Mognetti, Pietro Cicuta, and Lorenzo Di Michele. Programmable interactions with biomimetic dna linkers at fluid membranes and interfaces. Reports on progress in physics, 82(11):116601, 2019.
- [12] Stef AJ van der Meulen and Mirjam E Leunissen. Solid colloids with surface-mobile dna linkers. Journal of the American Chemical Society, 135(40):15129–15134, 2013.
- [13] Yin Zhang, Angus McMullen, Lea-Laetitia Pontani, Xiaojin He, Ruojie Sha, Nadrian C Seeman, Jasna Brujic, and Paul M Chaikin. Sequential self-assembly of dna functionalized droplets. Nature communications, 8(1):1–7, 2017.
- [14] Audray Harris, Giovanni Cardone, Dennis C Winkler, J Bernard Heymann, Matthew Brecher, Judith M White, and Alasdair C Steven. Influenza virus pleiomorphy characterized by cryo-electron tomography. Proceedings of the National Academy of Sciences, 103(50):19123–19127, 2006.
- [15] Francisco J Martinez-Veracoechea and Daan Frenkel. Designing super selectivity in multivalent nano-particle binding. Proceedings of the National Academy of Sciences, 108(27):10963–10968, 2011.

- [16] Galina V Dubacheva, Tine Curk, Bortolo M Moggetti, Rachel AuzeÅÅly-Velty, Daan Frenkel, and Ralf P Richter. Superselective targeting using multivalent polymers. Journal of the American Chemical Society, 136(5):1722–1725, 2014.
- [17] A. Wales. “Energy Landscapes”. Cambridge University Press, 2004.
- [18] T. Porstmann and S.T. Kiessig. “Enzyme immunoassay techniques An overview”. Elsevier, 1992.
- [19] S. Bhatia, D. Lauster, M. Bardua, K. Ludwig, S. Angioletti-Uberti, N. Popp, U. Hoffmann, F. Paulus, M. Budt, M. Stadtmüller, T. Wolff, A. Hamann, C. Böttcher, A. Herrmann, and R. Haag. Linear polysialoside outperforms dendritic analogs for inhibition of influenza virus infection in vitro and in vivo. Biomaterials, 138:22 – 34, 2017. ISSN 0142-9612. doi: <https://doi.org/10.1016/j.biomaterials.2017.05.028>. URL <http://www.sciencedirect.com/science/article/pii/S0142961217303460>.
- [20] Daniel Lauster, Maria Glanz, Markus Bardua, Kai Ludwig, Markus Hellmund, Ute Hoffmann, Alf Hamann, Christoph Böttcher, Rainer Haag, Christian PR Hackenberger, et al. Multivalent peptide–nanoparticle conjugates for influenza-virus inhibition. Angewandte Chemie International Edition, 56(21):5931–5936, 2017.

REVIEWER COMMENTS

Reviewer #1 (Remarks to the Author):

The theoretical part is well explained and the idea of range selectivity what the authors bring on the table is interesting. While the authors have given their arguments on both theory and experimental parts, the review was an opportunity for the authors to present their experimental data in a clear and convincing way, directly in the main text, figures, and supporting information instead of supplying raw data file. This would have convincingly validated the theory, at least for the systems like polymersomes with peptide ligand. However, I don't think this has been attempted, and the reasons are specified below,

Specific comments:

1. Figure 4b is a poor quality confocal fluorescence image. A representative confocal micrograph publishable in Nature Communication paper should have a good quality image, and a commercial Leica SP8 microscope, which is used for this article, should have been able to deliver so. In more Scientific terms, the image and figure caption does not give information about what a reader should look at the image regarding polymersomes and their surface coverage or adsorption probability. Also, there should have been two sets of images showing an obvious difference in the fluorescence signal from polymersomes of different ligand density. This is not even attempted. This would have satisfied me as a presentation of the experimental part of the work

2. The imaging experiments and corresponding analysis do not correctly represent the adsorption probability, because (1) overall fluorescence signal was evaluated and to note that, the signal comes from an added tag and not from the fluorescently-labelled ligand; (2) a good experiment would have been to do the imaging at single particle level as this is achievable by commercial confocal microscope. Single-particle images and corresponding image analysis will provide how many fluorescent polymersomes are attached on fixed cells. This is a quantity closer to the adsorption probability compared to the overall fluorescence signal which can be from free fluorescence molecules, polymerases with and without ligands. In short, the current fluorescence signal analysis does not represent true surface coverage of polymersomes.

3. Discussion in Page 17 in 2nd paragraph 'Even in its complete absence.....steric repulsion [14] might be enough'. This does not hold true for systems, for example, non-envelope viruses which has fixed ligands and no polymer brush for repulsion. Multiple groups have shown that for such viruses the adsorption probability (in experiments surface coverage or binding rate) increases non-linearly with increasing receptor concentration in lipid membrane and then reaches a plateau, but never decreases at a very high receptor concentration. So, the polymersome systems of this article cannot

be extended to such systems, and must be discussed and elaborated only for polymersomes type of systems.

Overall the idea of the theoretical work is to show that indeed the 'range-selectivity' idea is applicable for systems like polymersomes and other nanoparticles which are suitable for drug delivery, cancer therapy etc. Therefore, it is important that the authors take their time and perform thorough and systematic experiments to establish that indeed this is true for polymersomes.

Otherwise, it is difficult to evaluate the validity of the theory, particularly, for experimentalists who would wish to implement this knowledge to design nanoparticles with certain ligand density and/or target cells with receptor density that is within a certain range-selectivity. In the absence of any simulation and systematic experimental data, this article is not going to reach to the wider community of nanoparticle researchers who are interested in designing potent drug delivery agents and explore target cells.

Reviewer #2 (Remarks to the Author):

The authors have made extensive revisions to the manuscript, including provision of additional data, and addressed my questions with balance and clarity. The new version of the manuscript is substantially improved, and I have just a few minor comments that the authors may wish to consider to further increase clarity and appeal of the work for a general readership.

- I find the new title more appropriate, and appreciate the authors have taken this comment on board. The authors may also want to spell out in the manuscript which entropy they refer to in the title, as this may be misleading (see my interpretation in the first revision).

- In Fig. 4b, the appearance of the membrane stain suggests the not only the plasma membrane but also the nuclear membrane has been stained, indicating some permeability of the plasma membrane. It is known that fixation can cause cells to become permeable (as this process is not entirely non-perturbative to cell structure), and one may ask if this also enables polymersomes to enter into the cell. The author may want to carefully analyse their images to check for such an artefact. The polymersome staining for the cell in the top left of Fig. 4b, for example, appears to show a rim of enhanced intensity around what is likely to be the nucleus based on the cell membrane staining. This is hardly seen in the image as presented, but can be appreciated when enhancing the contrast for the polymersome stain.

- I understand the authors' argument that a linear x axis is preferable in Fig. 4a owing to the relatively small range of data being covered. Nonetheless, to facilitate comparison with the theoretical curves in Fig. 2, some readers may appreciate seeing these data additionally with a logarithmic axis. This could be easily implemented in the form of an inset to Fig. 4a, for example.

- In Fig. 4, I also noticed that a theoretical data point is missing for a normalised ligand density of 2. Please add it, or comment why it is missing.

- I appreciate the addition of a comparison between range selectivity and superselectivity in the revised manuscript. Along these lines, it would be interesting to see how sharply the binding range is defined in range selective systems. Have the authors considered implementing a parameter to quantify this, analogous to the superselectivity parameter α ? For the reader to be able to appreciate the steepness of the curves at the left and right boundaries of the binding range, it would be instructive to see the data in Fig. 2 additionally as a log-log plot. This could be implemented as a supplementary figure if deemed peripheral to the main paper.

- Page 16, the sentence "With some prominent exception [29, 30], this is true for most synthetic multivalent constructs, including functionalised colloids, nanoparticles or polymers" may be misleading. There are salient examples of synthetic multivalent constructs based on polymers where ligands are not in fixed positions. See references 10 and 11 in the manuscript, for example.

- Supplementary information, Fig. 1. I find it difficult to appreciate variations in size and homogeneity based on the 'correlograms' alone. Can the authors also present the size distribution as extracted from DLS data. A consistency check of the results obtained by DLS and TEM data would be appreciated. Also, how reproducible was the polymersome size from one batch to the next and was it stable over time? This should be checked unless all characterisation (DLS, TEM and cell binding) was performed on a single batch and within a very short time.

- Supplementary information, Fig. 2: I notice that the mean polymersome size varies appreciably across the different ligand densities. Whilst the authors acknowledge the relatively large dispersity in the main text, I suggest they also highlight that the variation in mean diameter is substantial (ranging from 20 and 53 nm). In any case, I am pleased to see this is considered in the fitting of the experimental data (Fig. 4).

- Supplementary Figs. 2 and 3: The authors may want to adopt consistent x axis ranges across the histograms shown on the right hand side to facilitate their comparison. Also, Fig. 2b lacks the PDI.

Reviewer #3 (Remarks to the Author):

While I usually study the reports of the other referees and the authors' replies to those as well, due to limited time availability I will restrict my new report to the changes the authors made in response to my comments.

The authors have addressed almost all my comments to my satisfaction. The exception is my request for an explanation of Equations 4 and 5. The response points to a new supplementary section, but that section seems to derive equation 7 and does not shed much light on equations 4 and 5. Perhaps the authors could expand the supplement a bit in order to also explain how those originate.

Apart from that, I recommend the manuscript for publication.

To all Reviewers,

we would like to thank you for the further comments we have received and for appreciating the extra efforts we have put in improving the first version of the paper. In the following, we will clarify the few additional concerns pointed out by the reviewers as well as highlight how we changed the main text in response to them (**these are highlighted in red**). Please note in this regard that **reference numbers used here, unless specifically stated otherwise, refer to the bibliography given below** at the end of this response letter, not to their numbers in the main manuscript.

Best,

Stefano Angioletti-Uberti (on behalf of all authors)

Reviewer 1 (Remarks to the Author):

R: The theoretical part is well explained and the idea of range selectivity what the authors bring on the table is interesting. While the authors have given their arguments on both theory and experimental parts, the review was an opportunity for the authors to present their experimental data in a clear and convincing way, directly in the main text, figures, and supporting information instead of supplying raw data file. This would have convincingly validated the theory, at least for the systems like polymersomes with peptide ligand. However, I don't think this has been attempted, and the reasons are specified below.

We thank the reviewer for acknowledging the relevance and importance of our work as well as the clarity of the theory behind it. It seems to us that the remaining concern is about the quality of our experimental analysis. We believe that the experimental characterisation we have done justifies our claim but clearly we need to improve its presentation as our efforts definitely do not fully transpire from the revised manuscript. We answer more specifically to their concerns below:

R: Figure 4b is a poor quality confocal fluorescence image. A representative confocal micrograph publishable in Nature Communication paper should have a good quality image, and a commercial Leica SP8 microscope, which is used for this article, should have been able to deliver so. In more Scientific terms, the image and figure caption does not give information about what a reader should look at the image regarding polymersomes and their surface coverage or adsorption probability. Also, there should have been two sets of images showing an obvious difference in the fluorescence signal from polymersomes of different ligand density. This is not even attempted. This would have satisfied me as a presentation of the experimental part of the work

We thank the reviewer for the comment. In response to this, and in order to also answer a comment from another reviewer, we have now replaced in the main text the old Fig.4b (now Fig.5 in the Supplementary Information) with a composite image including a 3D re-

construction rendering polymersome/membrane interaction. We did this because this more clearly shows both the quality of our imaging as well as the kind of experimental analysis we have performed to provide the adsorption data in Fig.4a. The new figure 4b and its caption have been changed accordingly (see Fig.1 here, Fig.4b in the main text).

Figure 1. Orthogonal projections and 3D reconstruction of CellMask Green stained cellular membrane after 1h of incubation with Cy5-polymersomes functionalized with 2% Angiopep ligand. Note how polymersomes only adsorb on the cellular membrane but they do not penetrate in the cell and no fluorescent signal is detectable within the nucleus. For this reason, we can exclude polymersomes penetration inside the cell up to the nucleus, at least up to the time of 1h after incubation with fixed cells, when the confocal microscopy experiments that measure adsorption were run. All confocal microscopy files from which experimental data have been calculated are available online in raw form from the DataFile.zip file.

The old Fig.4b instead has now been updated and moved to the Supplementary Information (Fig.5). We also present it here as Fig. 2 for ease of consultation for the reviewer.

As requested, this figure now presents a comparison between samples with different ligand loading, where the non-monotonic behaviour in the fluorescence signal is clearly visible.

Finally, we would also like to point out that we limited initially our experimental data presentation as in our opinion this effort is foremost theoretical and the experimental section, although important, represent a small section of the work. We however acknowledged the reviewer concerns and addressed their comments.

R: The imaging experiments and corresponding analysis do not correctly represent the adsorption probability, because (1) overall fluorescence signal was evaluated and to note that, the signal comes from an added tag and not from the fluorescently-labelled ligand; (2) a good experiment would have been to do the imaging at single particle level as this is achievable by commercial confocal microscope. Single-particle images and corresponding image analysis will provide how many fluorescent polymersomes are attached on fixed cells. This is a quantity closer to the adsorption probability compared to the overall fluorescence signal which can be from free fluorescence molecules, polymerases with and without ligands. In short, the current fluorescence signal analysis does not represent true surface coverage of polymersomes.

The fluorescent tag we add is stably inserted into the self-assembled structure of polymersomes and therefore useful for measuring polymersome binding to cellular membrane with and without ligands. We can exclude our fluorescence signal is associated to free fluorescence molecules or polymersomes without ligands for the following reasons. After the self-assembly, our samples undergo a dialysis step for over 24 h using a 3.5 kDa cut-off membrane to remove the organic solvent and any copolymer chain not assembled including Cy₅-PEG₁₁₃-PDPA₁₀₀ (fluorescently-labelled copolymer chains). Moreover, before every experiment, our samples are additionally purified by Size Exclusion Chromatography to homogenize the size distribution and remove any residual free molecule, including fluorescently-labelled copolymer chains. For what concerns the fluorescence signal coming from polymersomes without ligand, we believe the possibility of having polymersomes without ligands is extremely low. However, in our formulations, a polymersome sample with no

Figure 2. Confocal microscopy micrographs of polymersome binding to the cellular membrane after 1 h of incubation at 37°C. From left to right, Cy5-labelled polymersomes, CellMask Green-labelled cellular membrane of FaDu and colocalization of polymersomes and cellular membrane. From top to bottom, the series of images correspond to pristine (i.e., 0%, non-functionalised polymersomes), 1% and 10% Angiopep-2-functionalised polymersomes, respectively. A clear variation in the fluorescent signal can be seen among the samples from the Cy5-labelled polymersomes channel (in red). This variation is associated to the different amount of polymersomes adsorbed on the cellular membrane as a function of the Angiopep-2 functionalisation level. In particular, there is initially a clear increase in the fluorescent signal between 0% and 1%, which then drops again at 10% loading due to the non-monotonic adsorption behaviour, see also Fig.4a) in the main text.

ligand was included (Angiopep-2 0%). In this way we accounted for the behaviour of no ligand-polymersome and, as the reviewer can see from our results, the fluorescence derived from this sample is negligible (please observe Fig.5 in the supplementary Information, which is an updated version of the old Fig. 4b in the old manuscript). Furthermore, please note that fluorescently tagging the polymer instead of the ligand so that all formulations have the same amount of fluorescent dye independently of their ligand functionalisation is the best way to properly compare between samples. Indeed, it is the proper way to ensure that the differences in fluorescence intensity observed in confocal experiments for the different formulations are exclusively associated to the changing variable in them, i.e., the amount of ligands. Finally, we thank the reviewer for the suggestion of using single-particle tracking, but we believe this high-speed and high-sensitivity video microscopy techniques are more suitable for capturing fast dynamical process, or localize with nanometric precision nanoparticles within cellular sub-compartments. Here we are exclusively interested in measuring the total binding at the membrane level (please see the 3D rendering in Fig 4b in the new manuscript) and our experimental analysis has sufficient signal to measure it, providing us with a reliable estimation of the adsorption probability. For all these reasons, we think our experimental approach is reliable for the measurement we are proposing in this paper.

R: Discussion in Page 17 in 2nd paragraph ‘Even in its complete absence. steric repulsion [14] might be enough’. This does not hold true for systems, for example, non-envelope viruses which has fixed ligands and no polymer brush for repulsion. Multiple groups have shown that for such viruses the adsorption probability (in experiments surface coverage or binding rate) increases non-linearly with increasing receptor concentration in lipid membrane and then reaches a plateau, but never decreases at a very high receptor concentration. So, the polymersome systems of this article cannot be extended to such systems, and must be discussed and elaborated only for polymersomes type of systems.

We thank the reviewer for her/his comment and to give us the possibility to further clarify our model and conclusions. First, we do not make any absolute claim, in the sense that we never say that steric repulsion *must* be enough, it just *might* be, as stated in the text. Whether or not steric repulsion is enough depends on its relative strength with respect to

the binding interaction: the stronger it is, the higher the concentration of ligands where a drop in adsorption might be observed. Hence, the fact that one has not observed range selectivity in a specific system is not, *per se*, necessarily inconsistent with our theory. It would only be if a drop does not appear past the (ligand/receptor) density at which the drop is estimated to occur (within the uncertainties in using a coarse-grained model for its estimation). In other words, the fact that in an experiment one has not observed a drop in adsorption might simply be a result of not pushing the ligand/receptor density to high enough values. However, without an estimate of this critical density, it is impossible to say in absolute terms that no range selectivity has been observed. To show this, take for example the graphs we report in the parametric study, e.g. Fig.2c in the main paper. An experiment with the smallest receptor ($V_R = 50\text{nm}^3$) probing densities which would allow a maximum of 40 receptors to interact would never be able to observe range selectivity and one would instead see the “typical” monotonic binding. However, pushing the total number of receptors to higher values, or using a larger receptor to increase repulsion, would have instead shown the drop before the maximum of 40 receptors probed was reached. In this regard, although robust, range selectivity occurs in a range that depends on the system parameters and probing outside of it might give the wrong impression. Having said this, we do not expect in the counter-example provided by the referee to *ever* observe range-selectivity for the simple reason that receptors inserted in lipid membranes are, typically, *mobile*. We have scanned the literature to understand which studies the referee refers to and we have found the following: [1–4]. In all of them, this is indeed the case. It does not matter that the ligands on the virus are fixed, because as long as the receptors in the lipid membrane are mobile, as we argued in the paper when discussing the conditions for range selectivity, range selectivity is not expected to occur.

Finally, we would also like to point out another aspect. In biomimetic systems that measure the adsorption of a *naked* virus on a supported membrane, the steric interaction due to ligands and receptors only might be too small, even assuming fixed ligands and receptors, to observe range selectivity at experimentally achievable densities. However, the equivalent biological system they try to mimic is often different in an important and relevant way. The difference is that, in their natural form, various viruses including non-enveloped ones such as rotavirus [5] or adenovirus [6], are glycosylated. Similarly to the polymer brush on our polymersomes, these glycans can act as an additional source of (steric) repulsion. At the

same time, most host cells express a plethora of glycoproteins and proteoglycans that will act as a steric barrier to any nanoscopic object approaching (see our work in [7]). However, again this polymer layer is not (typically) included in supported lipid membranes used in laboratory experiments. For this reason, we feel it is important to suggest the *potential* relevance of range-selectivity for biological systems such as viruses attaching to the cell membrane, leaving the reader to decide, based on their specific system, whether or not this should be considered in the analysis of their data.

Reviewer 2 (Remarks to the Author):

R: The authors have made extensive revisions to the manuscript, including provision of additional data, and addressed my questions with balance and clarity. The new version of the manuscript is substantially improved, and I have just a few minor comments that the authors may wish to consider to further increase clarity and appeal of the work for a general readership.

We thank the referee for her/his appreciation of our further efforts. In the following, we provide an answer to the minor comments raised.

R: I find the new title more appropriate, and appreciate the authors have taken this comment on board. The authors may also want to spell out in the manuscript which entropy they refer to in the title, as this may be misleading (see my interpretation in the first revision).

We thanks the reviewer to suggest the title revision in the first place, which indeed makes things clearer. We believe further changing the new title to also specify what kind of entropy we mean would take away part of its appeal but we will discuss this point with the editor. In any case, we believe upon a full reading of the paper, it should all be clear.

R: In Fig. 4b, the appearance of the membrane stain suggests the not only the plasma membrane but also the nuclear membrane has been stained, indicating some permeability of the plasma membrane. It is known that fixation can cause cells to become permeable (as this process is not entirely non-perturbative to cell structure), and one may ask if this also enables polymersomes to enter into the cell. The author may want to carefully analyse their images to check for such an artefact. The polymersome staining for the cell in the top left of Fig. 4b, for example, appears to show a rim of enhanced intensity around what is likely to be the nucleus based on the cell membrane staining. This is hardly seen in the image as presented, but can be appreciated when enhancing the contrast for the polymersome stain.

We are aware that fixation process might alter the permeability of the cellular membrane. However, we can exclude polymersomes penetration inside the cell up to nucleus after 1h of incubation with fixed cells. As an example, we are showing here in Fig.3 (and now also in the revised Fig.4b in the main text) the orthogonal projections and 3D reconstruction of CellMask Green stained cellular membrane after 1h of incubation with Cy5-polymersomes functionalized with 2 % Angiopep ligand. The reviewer can appreciate how polymersomes only arrange on the cellular membrane whereas no fluorescent signal is detectable within the nucleus.

R: I understand the authors' argument that a linear x axis is preferable in Fig. 4a owing to the relatively small range of data being covered. Nonetheless, to facilitate comparison with the theoretical curves in Fig. 2, some readers may appreciate seeing these data additionally with a logarithmic axis. This could be easily implemented in the form of an inset to Fig. 4a, for example.

We have now implemented this change and present an inset with the logarithmic scale as suggested.

R: In Fig. 4, I also noticed that a theoretical data point is missing for a normalised ligand density of 2. Please add it, or comment why it is missing.

There was a mistake on our side, we should have actually removed that point altogether because we did not have a full characterisation of the size and polydispersity for that ligand loading (which is also required for providing a theoretical estimate). We have now removed also the experimental point from the figure to avoid confusion. Please note that we have also corrected the Table in the Supplementary Information, where this point was wrongly reported, as its data was actually for the point at 5% ligands loading (which was instead missing from the table and is now reported).

R: I appreciate the addition of a comparison between range selectivity and superselectivity in the revised manuscript. Along these lines, it would be interesting to see how sharply the binding range is defined in range selective

Figure 3. Orthogonal projections and 3D reconstruction of CellMask Green stained cellular membrane after 1h of incubation with Cy5-polymersomes functionalized with 2% Angiopep ligand. Note how polymersomes only adsorb on the cellular membrane but they do not penetrate in the cell and no fluorescent signal is detectable within the nucleus. For this reason, we can exclude polymersomes penetration inside the cell up to the nucleus, at least up to the time of 1h after incubation with fixed cells, when the confocal microscopy experiments that measure adsorption were run. All confocal microscopy files from which experimental data have been calculated are available online in raw form from the DataFile.zip file.

systems. Have the authors considered implementing a parameter to quantify this, analogous to the superselectivity parameter α ? For the reader to be able to appreciate the steepness of the curves at the left and right boundaries of the binding range, it would be instructive to see the data in Fig. 2 additionally as a log-log plot. This could be implemented as a supplementary figure if deemed peripheral to the main paper.

We have now added a graph in the Supplementary Information where we replot Fig.2 in a log-log scale so that evaluating the sharpness of the curve is easier, as suggested. On a related note, we want to highlight that indeed we have considered plotting the superselectivity parameter α for our case but realised that its definition is unsuitable for our system. More precisely, for α to be a good parameter, that is, a higher α should be corresponding to a sharper response, there are some conditions that are normally satisfied when considering typical multivalent binding but which do not occur when range-selectivity appears. In particular, because i) the binding probability monotonically drops to zero for high N_R , which means α keeps growing although the binding curve is essentially flat (this is because $\alpha = \frac{d\text{Log}\theta}{d\text{Log}N_R} = \frac{N_R}{\theta} \frac{d\theta}{dN_R}$, and θ in the denominator drops to zero, unlike for the typical case where $\theta \rightarrow 1$) and ii) α essentially corresponds to the exponent of the Hill curve used to fit $\theta(N_R)$, but in our case this functional form cannot be used because of our non-monotonic behaviour iii) α can be directly linked to the growth of the bound partition function and $\alpha > 1$ means super-linear behaviour, but no such connection can be made between super-linearity and the value of α in our case, again essentially because the partition function is non-monotonic in the number receptors. In this regard, an α -like parameter could be plotting simply $\text{abs}(\theta/dN_R)$, but unlike α there is now no connection to super-linearity and it would give an information that can be simply be gauged by simple visual inspection of the binding curves.

Page 16, the sentence “With some prominent exception [29, 30], this is true for most synthetic multivalent constructs, including functionalised colloids, nanoparticles or polymers” may be misleading. There are salient examples of synthetic multivalent constructs based on polymers where ligands are not in fixed positions. See references 10 and 11 in the manuscript, for example.

We believe there is a misunderstanding here. When we say “fixed”, we mean that the grafting point is fixed to a specific relative position on the construct. In ligand-functionalised polymers, e.g. those of references 10 and 11 cited by the referee, the ligands are fixed relative to their grafting sites. In this case, the ligands will necessarily move in absolute terms in space to follow any potential deformation of the backbone, but can at most swivel relative to their grafting point (exceptions in this regard are reversible supramolecular polymers that

can assemble and disassemble). In system with truly mobile ligands, see e.g. Ref [29,30] in the main manuscript) the ligands can move because they are not grafted covalently but simply adsorbed on the surface. This occurs because they are adsorbed via a cholesterol unit into a lipid bilayer, within which the cholesterol can move.

R: Supplementary information, Fig. 1. I find it difficult to appreciate variations in size and homogeneity based on the ‘correlograms’ alone. Can the authors also present the size distribution as extracted from DLS data. A consistency check of the results obtained by DLS and TEM data would be appreciated. Also, how reproducible was the polymersome size from one batch to the next and was it stable over time? This should be checked unless all characterisation (DLS, TEM and cell binding) was performed on a single batch and within a very short time.

As we specify in the main text, the correlograms are reported only as a proof that the colloidal suspension was stable but not to give any idea regarding their size and polydispersity. In fact, DLS can only provide an indirect measurement of size, that can be obtained only by using various additional assumptions, e.g. assuming negligible effects on light scattering and on the hydration shell due to the functionalisation with ligands, which are not correct in our system. For this reason, we provide TEM data which are a more direct and robust measurement of both size and polydispersity.

Our polymersomes are stable over time and their structure and size are very reproducible among batches. For this type of cell-binding mechanistic studies, we ensure a high level of control through purification techniques, including size exclusion chromatography, that allow us to control and separate precisely the size of interest, for reference please see [8, 9]. Finally, we also confirm that in any case all the formulations were prepared right before the experiments with cells and the confocal analysis, further limiting the potential for any aggregation to occur.

Supplementary information, Fig. 2: I notice that the mean polymersome size varies appreciably across the different ligand densities. Whilst the authors acknowledge the relatively large dispersity in the main text, I suggest they also highlight that the variation in mean diameter is substantial (ranging from 20

and 53 nm). In any case, I am pleased to see this is considered in the fitting of the experimental data (Fig. 4).

We thank the reviewer for her/his appreciation of our efforts to include polydispersity effects in the fitting. We are now also acknowledging the spread in size by adding the following sentence:

It should also be noted that our sample present a relatively large variation in the mean radius depending on the ligands loading, which varies between around 27 nm and 10 nm. Note that this variability in size distribution is taken into account in the fitting to the experimental data.

R: Supplementary Figs. 2 and 3: The authors may want to adopt consistent x axis ranges across the histograms shown on the right hand side to facilitate their comparison. Also, Fig. 2b lacks the PDI.

We have now corrected this as requested by the author and added the PDI in Fig.2b

Reviewer 3 (Remarks to the Author):

R: While I usually study the reports of the other referees and the authors' replies to those as well, due to limited time availability I will restrict my new report to the changes the authors made in response to my comments.

The authors have addressed almost all my comments to my satisfaction. The exception is my request for an explanation of Equations 4 and 5. The response points to a new supplementary section, but that section seems to derive equation 7 and does not shed much light on equations 4 and 5. Perhaps the authors could expand the supplement a bit in order to also explain how those originate.

Apart from that, I recommend the manuscript for publication.

We apologise, there was a misunderstanding in our first resubmission as we thought those were the equations whose derivation was requested. Having said that, we do not think it is necessary to derive Eq.4,5 here because these equations were not derived here for the first time. This was done by us in Ref.9 of the manuscript, where the details can be found. We now reword the manuscript to clarify this point and to point the reader to the original reference.

Old version: To provide a possible approximation, as in [9] we use a model built by combining previous results from Halperin [23] and Zhulina [24,25] to calculate the repulsive free-energy to insert an object in a polymer brush on a curved surface

New version: To provide a possible approximation, we use a model first derived in [9], built by combining previous results from Halperin [23] and Zhulina [24,25], to calculate the repulsive free-energy to insert an object in a polymer brush on a curved surface (details of the derivation can be found in the original paper, i.e. Ref.9).

[1] Nagma Parveen, Stephan Block, Vladimir P Zhdanov, Gustaf E Rydell, and Fredrik Hook. Detachment of membrane bound virions by competitive ligand binding induced receptor depletion.

- Langmuir, 33(16):4049–4056, 2017.
- [2] Marta Bally, Anders Gunnarsson, Lennart Svensson, Göran Larson, Vladimir P Zhdanov, and Fredrik Höök. Interaction of single viruslike particles with vesicles containing glycosphingolipids. Physical review letters, 107(18):188103, 2011.
- [3] Gustaf E Rydell, Andreas B Dahlin, Fredrik Höök, and Göran Larson. Qcm-d studies of human norovirus vlps binding to glycosphingolipids in supported lipid bilayers reveal strain-specific characteristics. Glycobiology, 19(11):1176–1184, 2009.
- [4] Daniele Di Iorio, Yao Lu, Joris Meulman, and Jurriaan Huskens. Recruitment of receptors at supported lipid bilayers promoted by the multivalent binding of ligand-modified unilamellar vesicles. Chemical Science, 11(12):3307–3315, 2020.
- [5] J Caust, ML Dyall-Smith, I Lazdins, and IH Holmes. Glycosylation, an important modifier of rotavirus antigenicity. Archives of virology, 96(3-4):123–134, 1987.
- [6] Gilles Cauet, Jean-Marc Strub, Emmanuelle Leize, Elsa Wagner, Alain Van Dorsselaer, and Monika Lusky. Identification of the glycosylation site of the adenovirus type 5 fiber protein. Biochemistry, 44(14):5453–5460, 2005.
- [7] Xiaohe Tian, Stefano Angioletti-Uberti, and Giuseppe Battaglia. On the design of precision nanomedicines. Science Advances, 2019.
- [8] Julia E Bartenstein, James Robertson, Giuseppe Battaglia, and Wuge H Briscoe. Stability of polymersomes prepared by size exclusion chromatography and extrusion. Colloids and Surfaces A: Physicochemical and Engineering Aspects, 506:739–746, 2016.
- [9] James D Robertson, Loris Rizzello, Milagros Avila-Olias, Jens Gaitzsch, Claudia Contini, Monika S Magoń, Stephen A Renshaw, and Giuseppe Battaglia. Purification of nanoparticles by size and shape. Scientific reports, 6(1):1–9, 2016.

REVIEWERS' COMMENTS:

Reviewer #1 (Remarks to the Author):

The authors have attempted to improve the experimental parts of the work and provided further experimental data and supporting discussion. I have minor comments that the authors may address

Question on the experimental Figures

1) What is the bright signal in the center of the two cells shown in XY plane of the micrograph, Figure 1?

The center of the cell is the nucleus and according to your experiment it should be dark as also shown in the Supporting confocal 2D micrograph

2) (i) As I see in Figure 1 the signal of the cell membrane (XY plane) is actually not limited to the cell membrane but also goes inside in the cytoplasm, mention where this is coming from

(It could possibly because of the signal comes from the adjacent 2D planes of the cells)

(ii) The signal from polymerosomes looks much dimmer in Figure 1 [scatter single (or 2-3 pixels) pixels] compared to the image shown in SI. Can add an intensity scale (or LUT), to have an idea of the signal intensity in all the images.

Reviewer #2 (Remarks to the Author):

The authors have addressed some of my comments in a satisfactory manner. However, other important comments have been rather poorly addressed, as described below.

1. I pointed out earlier that the authors chose to highlight the importance of entropy in the title. This is fine, and I do not suggest to change the title. However, if the authors want to attract a broad readership then they will wish to make it clearer in the abstract and the manuscript what type of entropy they refer to. For example, the authors refer to 'binding entropy' and 'avidity entropy' but these terms are not clearly defined in the manuscript and thus may be interpreted in different (and misleading) ways. Some clarification is required.

In their previous rebuttal letter, the authors stressed that the way the number of binding configurations grows in the different receptor density regimes is a key determinant of the effect observed. Following this notion, have the authors considered 'combinatorial entropy' or 'configurational entropy' as alternative, and possibly clearer, terminology?

2. The authors have revised Fig. 4b in light of comments by both reviewers. However, I was not convinced that the presented data support the author's claim that polymersomes do not penetrate into the cell. Firstly, the images are very small, making it difficult to appreciate the spatial distribution of fluorescence. Secondly, the combination of two colours in the orthogonal projections makes it difficult to appreciate the distribution of each individual colour. A set of three images, two with the individual colours plus one with merged colours would be helpful. Thirdly, I seem to appreciate some (albeit faint) red staining inside the nucleus. Fourthly, the images shown do not permit to assess if polymersomes enter the cytosol.

I recommend the authors also make it clear in the text that their membrane stain also stains membranes other than the plasma membrane. Moreover, the 3D reconstructions do not seem to add new information beyond those contained in the orthogonal projections, and may thus be omitted if space is limited.

3. With regard to my previous comment related to the sentence "With some prominent exception [29, 30], this is true for most synthetic multivalent constructs, including functionalised colloids, nanoparticles or polymers" on page 16. It is not clear why the authors have not sought to address this comment in the manuscript. If a reviewer misunderstands, then future readers may, too.

Moreover, I do not understand the argument made by the authors in the rebuttal letter. For the physics of the problem it appears to be irrelevant if the ligands are covalently attached to a flexible backbone or if the ligands are free to diffuse on a rigid support. Configurational flexibility is provided in both cases and one may even argue it is higher with a flexible polymer backbone (where ligands are free to re-organise in 3D) as compared to a membrane on a particle surface (where ligands are confined to a 2D space).

4. I have substantial reservations with regard to the use of TEM as the only method for particle sizing, particularly so because details on the TEM sample preparation were not provided in the manuscript. The TEM micrographs shown in Supporting Figures 2 and 3 suggest the samples were dried. Artefacts of sample drying are well documented to affect the apparent size and shape of liposomes, and I suspect that polymersomes may suffer similar issues. Moreover, preferential binding to the grid may entail a skewed representation of the particle size distributions in solution.

I was expecting the DLS data to instill additional confidence. This is a well-established technique for particle sizing, and I do not understand the authors' arguments with regard to limitations of this technique: light scattering is the very basis of the technique, and the hydration shell should ideally

be included in the consideration of particle size because it is the swollen particle (rather than the dried particle as seen by TEM) that interacts with the cell membrane.

5. Last but not least, the authors claim in the last rebuttal letter that their polymersomes are stable over time and their structure and size are very reproducible among batches. However, they fail to provide evidence for this crucial experimental aspect.

To all Reviewers,

we would like to thank you for the further comments we have received. In the following, we will clarify the few points raised by the reviewers as well as highlight how we changed the main text in response to them (all highlighted in red).

Note: For self-completeness in each response, some answers and the accompanying data are reported twice

Best,

Stefano Angioletti-Uberti (on behalf of all authors)

Reviewer 1 (Remarks to the Author):

The authors have attempted to improve the experimental parts of the work and provided further experimental data and supporting discussion. I have minor comments that the authors may address.

Question on the experimental Figures. What is the bright signal in the center of the two cells shown in XY plane of the micrograph, Figure 1? The center of the cell is the nucleus and according to your experiment it should be dark as also shown in the Supporting confocal 2D micrograph

As the reviewer suggests, it might well be that the CellMaskTM Plasma Membrane Stain might leak into the nuclear membranes, however the most likely explanation is that both cells were fixed during their mitosis with the consequent condensation of the membranous compartments highlighting the nuclear shape. We provided all the data and figures used for the binding assay and as it can be appreciated the CellMask stains the membrane and leaves the nucleus dark in most cases. It is paramount to point out that independently that the cell nuclei are stained or not, the CellMaskTM marker is used to identify the cell membrane and use it as reference for our analysis, focused on the polymersomes signal. Here the fluorescence is mostly showed around the cell borders and it is not internalised nor is found in the nuclei as showed by the orthonormal XZ and YZ sections. In general, we do recognise the overall difficulty to clearly identify the cellular membrane in Fig. 4b, and consequently, even assessing the exact location of polymersomes might be challenging and misleading. For this reason, and after considering the comments of both reviewers concerning Fig. 4b, we have realised that Fig. 4b might not be completely representative of our results and might have lead to misinterpret the results we are presenting. For this reason:

(1) we now provide additional images where less of this phenomenon (i.e., CellMask staining within the cell) is observed (see revised Figure 4 in the main text). As reviewers 1 and 2 suggested, we present them as set of three images where the single channels and an intensity scale are included.

(2) For consistency and as an internal check, in order to avoid any confusion between the confocal images shown in this work and the way the analysis of the experimental data has been carried out, we have re-measured the adsorption probability as a function of the ligands

number removing from the set of experimental data those confocal images where staining within the cell seems to appear. As a result, the new polymersomes adsorption values (see for reference Table 1 and Figure 1 in this response letter) do not significantly differ from the original one (5% variation in the worst case), thus proving that potential difficulties in identifying the exact position of the cell membrane do not affect any of our conclusions. Finally, we have amended (adding the sentence here in red) a sentence in the Methods section to commenting on the way our analysis was done:

“Every stack could include more than one cell and the fluorescence measurement per formulation was carried out on a minimum number of 40 cells. *Cells that were fixed during mitotic events were discarded from the analysis*”

Ligands load	0%	0.5%	1.0%	5.0%	10.0%
Adsorption (un-normalised), old	4274766	4826184	7939927	5751906	4114895
Adsorption (un-normalised), revised	4274766	4826184	7921366	5832617	4348228
$\Delta (\text{Revised} - \text{Old}) / \text{Revised}$	0.0 %	0.0 %	0.2 %	1.3 %	5.3 %

Table 1: Old vs revised adsorption values (see main text of rebuttal for details). The largest difference is, at most, $\approx 5\%$, irrelevant for our conclusions.

(i) As I see in Figure 1 the signal of the cell membrane (XY plane) is actually not limited to the cell membrane but also goes inside in the cytoplasm, mention where this is coming from (It could possibly because of the signal comes from the adjacent 2D planes of the cells) (ii) The signal from polymerosomes looks much dimmer in Figure 1 [scatter single (or 2-3 pixels) pixels] compared to the image shown in SI. Can add an intensity scale (or LUT), to have an idea of the signal intensity in all the images.

(i) We are aware that after a certain period of time after staining, the staining agent can be internalised into the cell and that fixation processes can favour this event. Specifically in Fig. 4b this might have happened and that is the reason why the nucleus looks very bright. However, it is usually the case for this staining agent for the cellular membrane (CellMaskTM

Figure 1: Old and revised values adsorption values (excluding cells with mitotic events, see rebuttal letter for details) for the measured experimental adsorption.

Green), under confocal microscopy and under the optimised experimental conditions, that the membrane is not imaged like just a contour line but seems to cover the whole cell surface. We are aware that this might lead to think that the staining agent goes into the cytoplasm, but we believe this behaviour might be a peculiar property of this staining agent as also visible from the promotional images provided from the manufacturer (please have a look here for more details <https://www.thermofisher.com/order/catalog/product/C37608/C37608>).

(ii) We have now added a LUT in all the images presented as the reviewer suggested.

Reviewer 2 (Remarks to the Author):

The authors have addressed some of my comments in a satisfactory manner. However, other important comments have been rather poorly addressed, as described below.

I pointed out earlier that the authors chose to highlight the importance of entropy in the title. This is fine, and I do not suggest to change the title. However, if the authors want to attract a broad readership then they will wish to make it clearer in the abstract and the manuscript what type of entropy they refer to. For example, the authors refer to ‘binding entropy’ and ‘avidity entropy’ but these terms are not clearly defined in the manuscript and thus may be interpreted in different (and misleading) ways. Some clarification is required. In their previous rebuttal letter, the authors stressed that the way the number of binding configurations grows in the different receptor density regimes is a key determinant of the effect observed. Following this notion, have the authors considered ‘combinatorial entropy’ or ‘configurational entropy’ as alternative, and possibly clearer, terminology?

We had used the terms “binding entropy” and “avidity entropy” without defining them in the manuscript because these are part of the technical jargon on multivalent interactions. However, we agree with the referee that to make the manuscript clearer and also more accessible to a wider audience, we can do better. In this regard, the suggestion of the referee is perfect and we now use use the term “combinatorial entropy” as suggested, as this perfectly pinpoints the type of entropy we are referring to.

Also, please note that in order to make it compliant with editorial policies, the title has now changed again.

The authors have revised Fig. 4b in light of comments by both reviewers. However, I was not convinced that the presented data support the author’s claim that polymersomes do not penetrate into the cell. Firstly, the images are very small, making it difficult to appreciate the spatial distribution of fluorescence.

Secondly, the combination of two colours in the orthogonal projections makes it difficult to appreciate the distribution of each individual colour. A set of three images, two with the individual colours plus one with merged colours would be helpful. Thirdly, I seem to appreciate some (albeit faint) red staining inside the nucleus. Fourthly, the images shown do not permit to assess if polymersomes enter the cytosol. I recommend the authors also make it clear in the text that their membrane stain also stains membranes other than the plasma membrane.

We thank the reviewer for the annotation and we agree with the reviewer that in Fig. 4b the nucleus might be stained as well in a way that it results almost brighter than the cellular membrane. However, for this particular point we want to add few considerations.

- We are aware that after a certain period of time, the staining agent can be internalised into the cell and that fixation processes can favour this event. Specifically in Fig. 4b this might have happened and that is the reason why the nucleus looks very bright. However, it is usually the case for this staining agent for the cellular membrane (CellMask™ Green), under confocal microscopy and under the optimised experimental conditions, that the membrane is not imaged like just a contour line but seems to cover the whole cell surface. We are aware that this might lead to think that the staining agent goes into the cytoplasm and consequently that polymersomes are localised in the cytoplasm, as the reviewer has pointed out. However, we believe this behaviour might be a peculiar property of this staining agent as also visible from the promotional images provided from the manufacturer (please have a look here for more details <https://www.thermofisher.com/order/catalog/product/C37608/C37608>).

- The cells depicted in Fig. 4b are close to mitosis events, during which the cellular membrane morphology changes. In fact, many cells round up during mitosis and the cell membrane can be localised very close to the nucleus. This re-arrangement would make difficult to clearly discriminate the nucleus from the cell membrane particularly in cases where the nucleus is not separately stained with typical cell nucleus staining such as DAPI or Hoechst.

In general, we do recognise the overall difficulty to clearly identify the cellular membrane in Fig. 4b, and consequently, even assessing the exact location of polymersomes might be challenging and misleading.

For this reason, and after considering the comments of both reviewers concerning Fig. 4b,

we have realised that Fig. 4b might not be completely representative of our results and might have lead to misinterpret the results we are presenting. For this reason:

(1) we now provide additional images where less of this phenomenon (i.e., CellMask staining within the cell) is observed (see revised Figure 4 in the main text). As reviewers 1 and 2 suggested, we present them as set of three images where the single channels and an intensity scale are included.

(2) For consistency and as an internal check, in order to avoid any confusion between the confocal images shown in this work and the way the analysis of the experimental data has been carried out, we have re-measured the adsorption probability as a function of the ligands number removing from the set of experimental data those confocal images where staining within the cell seems to appear. As a result, the new polymersomes adsorption values (see for reference Figure 1 in this response letter) do not significantly differ from the original one ($\approx 5\%$ variation in the worst case), thus proving that potential difficulties in identifying the exact position of the cell membrane do not affect any of our conclusions.

Finally, we have amended (adding the sentence here in red) a sentence in the Methods section to comment on the way our analysis was done:

“Every stack could include more than one cell and the fluorescence measurement per formulation was carried out on a minimum number of 40 cells. *Cells that were fixed during mitotic events were discarded from the analysis*”

Ligands load	0%	0.5%	1.0%	5.0%	10.0%
Adsorption (un-normalised), old	4274766	4826184	7939927	5751906	4114895
Adsorption (un-normalised), revised	4274766	4826184	7921366	5832617	4348228
$\Delta (\text{Revised} - \text{Old}) / \text{Revised}$	0.0 %	0.0 %	0.2 %	1.3 %	5.3 %

Table 1: Old vs revised adsorption values (see main text of rebuttal for details). The largest difference is, at most, $\approx 5\%$, irrelevant for our conclusions.

Moreover, the 3D reconstructions do not seem to add new information beyond those contained in the orthogonal projections, and may thus be omitted if space is limited.

Figure 1: Old and revised values adsorption values (excluding cells with mitotic events, see rebuttal letter for details) for the measured experimental adsorption.

We followed the reviewer suggestion and have now removed the 3D reconstruction from the new version.

With regard to my previous comment related to the sentence “With some prominent exception [29, 30], this is true for most synthetic multivalent constructs, including functionalised colloids, nanoparticles or polymers” on page 16. It is not clear why the authors have not sought to address this comment in the manuscript. If a reviewer misunderstands, then future readers may, too. Moreover, I do not understand the argument made by the authors in the rebuttal letter. For the physics of the problem it appears to be irrelevant if the ligands are covalently attached to a flexible backbone or if the ligands are free to diffuse on a rigid support. Configurational flexibility is provided in both cases and one may even argue it is higher with a flexible polymer backbone (where

ligands are free to re-organise in 3D) as compared to a membrane on a particle surface (where ligands are confined to a 2D space).

We have now addressed this in the main text when discussing the effect of mobility. In particular, we have added the part in red below:

“When receptor (or ligand) mobility is high and can induce local changes of their grafting density, instead, a theoretical analysis of the binding mechanism should include its effects. In particular, this is also the case for functionalised polymers (see e.g. Ref.10,11), where the backbone to which ligands are attached can deform and thus change the local ligand density (albeit with an associated configurational entropy penalty). Whereas a full description goes beyond the scope of the present manuscript, we notice that this could be done using some recent derived by Mognetti et al in [Mognetti, Cicutta, Di Michele, “Programmable interactions with biomimetic dna linkers at fluid membranes and interfaces. Reports on progress in physics” 82(11):116601 (2019); Robin De Gernier, Tine Curk, Galina V Dubacheva, Ralf P Richter, and Bortolo M Mognetti “A new configurational bias scheme for sampling supramolecular structures”, The Journal of chemical physics, 141(24):244909 (2014)]”.

Please notice that we have added the second reference describing a computational techniques specifically developed to consider the effect of backbone re-arrangement in determining the binding of ligand-functionalised polymers.

Just as a short explanation regarding the second point, the referee is correct in saying the polymer backbone can reconfigure - and thus ligands can move - compared to a rigid substrate with grafted ligands. However, unlike the case of mobile ligands on a 2D rigid substrate, the backbone reconfiguration does not come for free but is instead associated with an entropic penalty. In other words, one cannot just move a ligand keeping everything else fixed (as in the 2D rigid case) but instead moving a ligand means imposing a deformation to the whole backbone. As we discussed also previously in the text, a detailed analysis of these effects is outside the scope of this work, but can be done with the cited techniques.

I have substantial reservations with regard to the use of TEM as the only method for particle sizing, particularly so because details on the TEM sample preparation were not provided in the manuscript. The TE micrographs shown in Supporting Figures 2 and 3 suggest the samples were dried. Artefacts of sample

drying are well documented to affect the apparent size and shape of liposomes, and I suspect that polymersomes may suffer similar issues. Moreover, preferential binding to the grid may entail a skewed representation of the particle size distributions in solution. I was expecting the DLS data to instill additional confidence. This is a well-established technique for particle sizing, and I do not understand the authors' arguments with regard to limitations of this technique: light scattering is the very basis of the technique, and the hydration shell should ideally be included in the consideration of particle size because it is the swollen particle (rather than the dried particle as seen by TEM) that interacts with the cell membrane.

Note: This point was addressed in an informal email exchange with the editor and a provisional answer was already provided, which is summarised here.

As requested by the reviewer, we now provide DLS data in the Supplementary files (Supplementary Figure 2, corresponding to Figure 3 here in the rebuttal letter). In particular we submit the number distribution for all the samples used in the experiments, which shows that the samples display a narrow and well-defined size distribution. However, as can be seen, there is some discrepancy between the TEM and DLS size distributions, with DLS yielding larger sizes than TEM. The smaller size observed in TEM could be partly due to the drying effects suggested by the reviewer.

Please, also note that:

(i) Our samples were fluorescently-labelled with Cy5 dye, whose excitation wavelength is 647 nm, that is, almost coincident with the wavelength of the laser beam (633 nm) of the Zetasizer machine. Therefore, the resulting fluorescence from the samples is also collected in the detector, which can add a source of error in the estimation of size. (ii) DLS measures the hydrodynamic radius, that is, the radius of the particle plus the hydration shell. Because this hydration shell is of the orders of a few nanometers, for large particles (such as 100 nm-vesicles) the ratio of the hydrodynamic radius/the true particle radius is basically one and DLS provides a good estimate of particle size. However, in our samples many of the particles observed in TEM correspond to small micellar-like structures, with size of the order of 20-30 nanometers, and hence DLS does not provide a reliable estimate of their real size.

In any case and because both techniques have their limitations, again as suggested by the reviewer, we have now considered DLS and TEM data as upper and lower limits of the particle size in our theory and re-fitted Fig. 4. As it can be appreciated, there is little difference between the two sets. The values of the fitting parameters remain within a fraction from each other, well within the limitations of using a coarse-grained model for describing the binding probability which we only expect to be correct for an order-of-magnitude estimate. More importantly, the non-monotonicity of the binding curve which we dubbed range selectivity is not affected. This is because range selectivity is a robust phenomenon, that is, is not strongly dependent on the exact details of the parameters describing the system such as the particle size distribution. In fact, its occurrence arises simply from the general physics of ligand-receptor interactions, in particular, on the scaling obeyed by the different interactions as discussed in the text. As suggested by the reviewer we have now added this point in the manuscript to make this conclusion even clearer and point out the robustness of the phenomenon:

Furthermore, it should be noted that despite using two estimates for the particle size distribution, shown as the two sets of theoretical data in Fig. 4 (see details in the Methods Section), the theoretical model still predicts the non-monotonic behaviour observed in experiments. This observation corroborates the fact that range selectivity is a robust phenomenon, not significantly affected by the system polydispersity. In fact, this robustness should be expected, given that the occurrence of range selectivity only depends on very mild assumptions on the scaling of the repulsive interactions.

Last but not least, the authors claim in the last rebuttal letter that their polymersomes are stable over time and their structure and size are very reproducible among batches. However, they fail to provide evidence for this crucial experimental aspect.

We thank the reviewer for having pointed this out and we are more than keen to provide support to our claim hoping to help the reviewer clarify his/her doubts. Please find here attached in Figure 2 additional DLS measurement of different samples prepared among different batches (also reported as a new section in the Supplementary Information). Specifically, in panel A “Time stability” we report the DLS number distribution of the samples used for

this study at the time when the experiments have been performed (AP-PEGPDPA psomes $t=0$, blue solid lines) and 1 year later (AP-PEGPDPA psomes 1 year, blue dashed lines). Unfortunately, we run out of the samples PEG-PDPA 0.5% AP and PEG-PDPA 5% AP. In panel B “Batch reproducibility”, we report the DLS characterisation of the samples used for the experiments in this work (AP-PEGPDPA psomes $t=0$, blue solid lines) and a second independent batch of ligand-and fluorophore-conjugated polymersomes immediately after the sample preparation (AP-PEGPDPA independent batch, green solid lines). In order to prove that the presence of the ligands or the fluorophore in our polymersome formulations does not affect their size, in panel C “Batch reproducibility among pristine and functionalized psomes” we report a comparison between the number distribution of the ligand- and fluorophore-conjugated polymersomes used for the experiments in this study (AP-PEGPDPA psomes $t=0$, blue solid lines) and pristine PEG-PDPA polymersomes immediately after the preparation (pristine psomes $t=0$, black solid line) and 1 year after their preparation (pristine psomes 1 year, red solid line). As the reviewer can appreciate, no significant differences can be observed among different batches and over long periods of time.

A. Time stability

B. Batch reproducibility

C. Batch reproducibility among pristine and functionalized psomes

Figure 2: Study of the time-stability of our polymersomes via DLS, see details in the rebuttal letter.